# DDRP: Real-time phenology and climatic suitability modeling of invasive insects

**Brittany S. Barker**[1,2]*, **Leonard Coop**[1,2], **Tyson Wepprich**[3], **Fritzi Grevstad**[3], **Gericke Cook**[4]

**1** Oregon IPM Center, Oregon State University, Corvallis, OR, United States of America, **2** Department of Horticulture, Oregon State University, Corvallis, OR, United States of America, **3** Department of Botany and Plant Pathology, Oregon State University, Corvallis, OR, United States of America, **4** USDA Animal and Plant Health Inspection Service, Fort Collins, CO, United States of America

* brittany.barker@oregonstate.edu

**Data Availability Statement:** On GitHub, we have uploaded the most current version of DDRP along with a user guide and species parameter files (github.com/bbarker505/ddrp_v2), Perl and Octave scripts used for temporal downscaling of monthly

## Abstract

Rapidly detecting and responding to new invasive species and the spread of those that are already established is essential for reducing their potential threat to food production, the economy, and the environment. We describe a new spatial modeling platform that integrates mapping of phenology and climatic suitability in real-time to provide timely and comprehensive guidance for stakeholders needing to know both where and when invasive insect species could potentially invade the conterminous United States. The Degree-Days, Risk, and Phenological event mapping (DDRP) platform serves as an open-source and relatively easy-to-parameterize decision support tool to help detect new invasive threats, schedule monitoring and management actions, optimize biological control, and predict potential impacts on agricultural production. DDRP uses a process-based modeling approach in which degree-days and temperature stress are calculated daily and accumulate over time to model phenology and climatic suitability, respectively. Outputs include predictions of the number of completed generations, life stages present, dates of phenological events, and climatically suitable areas based on two levels of climate stress. Species parameter values can be derived from laboratory and field studies or estimated through an additional modeling step. DDRP is written entirely in R, making it flexible and extensible, and capitalizes on multiple R packages to generate gridded and graphical outputs. We illustrate the DDRP modeling platform and the process of model parameterization using two invasive insect species as example threats to United States agriculture: the light brown apple moth, *Epiphyas postvittana*, and the small tomato borer, *Neoleucinodes elegantalis*. We then discuss example applications of DDRP as a decision support tool, review its potential limitations and sources of model error, and outline some ideas for future improvements to the platform.

## Introduction

Invasive insects in the United States are a significant threat to the economy, environment, food security, and human health [1–3]. They cause billions of dollars in damage to forests each

climate averages to daily averages (github.com/bbarker505/dailynorms), and temporally downscaled climate data (github.com/bbarker505/1990_daily_30yr). On Zenodo, we have archived the version of DDRP used for this study (https://doi.org/10.5281/zenodo.3832731), and the scripts and outputs of the temporal downscaling analysis of climate data (https://doi.org/10.5281/zenodo.3601671 and https://doi.org/10.5281/zenodo.3879627, respectively).

**Funding:** This work was funded by the USDA APHIS PPQ Cooperative Agricultural Pest Survey (CAPS) grant no. AP18PPQHQ000C009 (Cooperative Agreement) to L.C. (URL: https://www.aphis.usda.gov/aphis/), the Center for Plant Health Science and Technology (CPHST) program grant no. 15-8130-0304-CA to L.C., the Applied Research and Development Program (NIFA-CPPM-ARDP) grant no. 2014-07773 to L.C. (https://nifa.usda.gov/funding-opportunity/crop-protection-and-pest-management), and the Department of Defense Strategic Environmental Research and Development Program grant no. W912HQ-17-C-0051 (project RC-2701) to F.G. and L.C. (https://www.serdp-estcp.org/). The funders had no role in study design, data collection and analysis, decision to publish, or preparation of the manuscript.

**Competing interests:** The authors have declared that no competing interests exist.

year [1, 2], and their potential cost to food crop production is among the highest of any country [3]. Insect invasions in the United States also reduce the abundance and diversity of native species, which negatively impacts ecosystem functions and services such as soil health, nutrient cycling, and wildlife habitat [1, 2]. Rapidly detecting and responding to new invasive insects and the spread of those that are already established before they can cause significant economic and environmental damage has therefore become a major priority [2, 4].

Modeling climatic suitability (risk of establishment) and the timing of seasonal activities (phenology) of invasive insect species can help stakeholders including farmers, natural resource managers, and surveillance teams detect and prevent their establishment, slow their spread, and manage existing populations more sustainably and economically [5]. Estimates of climatic suitability identify areas to concentrate surveillance or management resources and efforts [6, 7], whereas real-time (i.e. current) or forecasted predictions of phenology can improve the timing of surveillance and integrated pest management (IPM) efforts such as monitoring device installation, pesticide applications, and biological control release [8–10]. Additionally, estimates of climatic suitability, phenology, and voltinism (number of generations per year) can help growers predict the impact of pests and diseases on agricultural production and associated economic losses [11].

Degree-day models that predict insect phenology are an established tool for decision support systems that assist stakeholders with scheduling surveillance, monitoring or IPM operations for numerous pest species over the growing season [12–15]. Most degree-day models predict phenology by measuring linear relationships between temperature and development rate, and they employ daily time steps to estimate degree-days using daily minimum and maximum temperature ($T_{min}$ and $T_{max}$, respectively) data. In the daily time step, degree-days accumulate if heat exceeds the lower developmental temperature threshold of a species (and below its upper threshold for some calculation methods) during a 24-hour period [12, 13, 15]. Several web-based platforms host degree-day models for insect pest species in the United States, offering users opportunities to model phenology of multiple species at single locations (site-based model) or across a certain area (spatialized model). These platforms include but are not limited to Michigan State University's Enviroweather (https://www.enviroweather.msu.edu), Oregon State University's USPEST.ORG (https://uspest.org/wea/), the Spatial Analytic Framework for Advance Risk Information System (SAFARIS; https://safaris.cipm.info/), and the USA National Phenology Network (https://www.usanpn.org) [8, 16].

Despite their widespread use, currently available degree-day modeling platforms are in need of improvements. None of them integrate predictions of phenology and climatic suitability, which would provide guidance on the question of both where *and* when—e.g. is an area at high risk of establishment, and if so, then when will the species emerge or begin a specific activity? For most species, addressing this two-part question would require finding, potentially purchasing, and learning how to use two separate platforms. Additionally, many phenology modeling platforms use oversimplified models that make broad assumptions about insect biology, such as assuming a single lower developmental temperature threshold for multiple species, or assuming that an entire population emerges from overwintering at a single time. However, developmental temperature thresholds may vary widely across insect species, and development rates often vary within populations [17–19]. A biologically unrealistic model may produce inaccurate predictions of phenological events (e.g. spring emergence, first adult flight, egg-hatching) or voltinism. Moreover, most platforms are capable of forecasting phenology only a week or two into the future in specific states or regions. However, stakeholders may need to plan operations several weeks in advance, potentially in areas that are outside the geographic bounds of predictive models.

In this study, we introduce a new spatial modeling platform, DDRP (short for Degree-Days, establishment Risk, and Phenological event maps) that generates real-time and forecast predictions of phenology and climatic suitability (risk of establishment) of invasive insect species in the conterminous United States (CONUS). The objective of DDRP is to provide a multi-species modeling tool that can improve the efficiency and effectiveness of programs that aim to detect new or spreading invasive insect species in the United States, to monitor and manage species such as IPM insect pests that are already well-established, and to improve programs for classical biological control insects. The platform is written entirely in the R statistical programming language [20], making it flexible and extensible, and has a simple command-line interface that can be readily implemented for online use. Gridded temperature data for DDRP may include the entire CONUS or a specific region or state, and may be at any spatial resolution that can be handled by the user's computing system. DDRP will generally use observed and future (forecast or recent average) temperature data because it was designed to be run as a within-season decision support tool that can provide guidance on where and when to expect the pest to appear each year, but it will accept temperature data for any time period. Model outputs include gridded (raster) and graphical (map) outputs of life stages present, number of generations, phenological events, and climatic suitability.

First, we describe the modeling process and workflow of DDRP, summarize types of model outputs, and review its system and software requirements. Next, we demonstrate its capabilities and the process of model parameterization using two invasive insect species which threaten agricultural biosecurity in the United States: the small tomato borer, *Neoleucinodes elegantalis* [Guenée (Lepidoptera: Crambidae)], and the light brown apple moth, *Epiphyas postvittana* [Walker, 1863 (Lepidoptera: Tortricidae)]. These species were chosen because they have been well-studied in terms of their developmental requirements, and previous climatic suitability studies provide a basis for parameterizing the climatic suitability model in DDRP. Additionally, models for these species are intended to aid surveillance teams at the Cooperative Agricultural Pest Survey (CAPS) pest detection program, which supports the USDA Animal and Plant Health Inspection Service (APHIS) as it works to safeguard agricultural and environmental resources in the United States. We used population monitoring data for *E. postvittana* in California to test the hypothesis that DDRP can correctly predict the timing of first spring egg laying and the generation length of the species. Additionally, we used a validation data set consisting of presence localities from California (*E. postvittana*) and Brazil (*N. elegantalis*) to test the hypothesis that DDRP can correctly predict each species' known distribution in these areas. The DDRP platform will be a useful decision support tool for preventing, monitoring, and managing new and existing invasive pests of agriculture and natural resources in the United States.

## Methods

### 1) Model inputs

**Temperature data.** DDRP requires daily $T_{min}$ and $T_{max}$ data in a gridded format for an area of interest in CONUS (Fig 1). For real-time modeling, we have been using daily $T_{min}$ and $T_{max}$ data at a 4 km spatial resolution from the PRISM (Parameter-elevation Relationships on Independent Slopes Model) database (available at https://prism.oregonstate.edu) [21]. Daily PRISM data become available *ca.* 1 day after weather station observations are reported, and are typically updated and improve in quality as more observations are added (see PRISM website for details). The phenology mapping system of the USA National Phenology Network [8] uses Real-Time Mesoscale Analysis (RTMA) weather data at a 2.5 km resolution, which are available within hours after data are observed. The daily $T_{min}$ and $T_{max}$ RTMA data set could

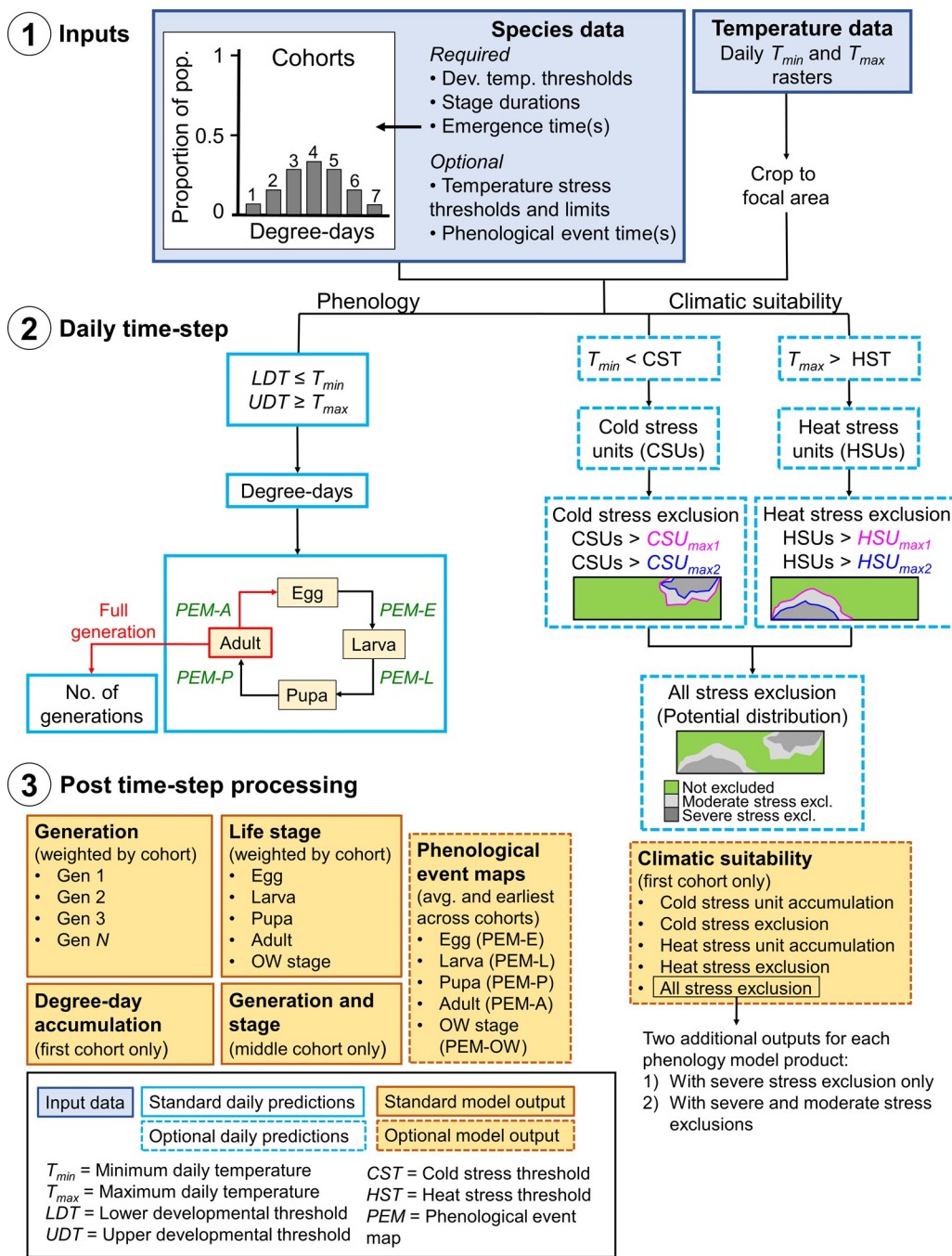

**Fig 1. Schematic of the DDRP model framework.** 1) Input data sets (blue shaded boxes) include a) data on the developmental requirements, climatic tolerances (optional), and emergence times of population cohorts of a species (Table 1), and b) daily minimum and maximum temperature ($T_{min}$ and $T_{max}$, respectively) data. 2) Hollow blue boxes indicate calculations conducted on each daily time step, where a dashed outline represents calculations for climatic suitability. Phenological event map (PEM) calculations for each life stage (E = egg, L = larva, P = pupa, A = adult) are shown in green font. A full generation is counted when adults lay eggs (in red), and the number of generations subsequently increases. 3) After the daily time step completes, DDRP combines the results across all cohorts and exports the model outputs as multi-layer raster (".tif") and summary map (".png") files (orange shaded boxes). Orange shaded boxes with a dashed line represent model outputs for PEMs and climatic suitability.

potentially be used in DDRP; however, the RTMA methodology lacks PRISM's update and quality control regimes [21]. Another alternative is Daymet v3, which offers daily climate data for North America, Hawaii, and Puerto Rico at a very high spatial resolution (1 km) (https://daymet.ornl.gov) [22]. However, Daymet data are released months after the end of each year, so they would be less useful for within-season modeling and decision support.

For forecast modeling, DDRP is currently configured to use either monthly-updated, daily-downscaled NMME (North American Multi-Model Ensemble) 7-month forecasts at a 4 km resolution [23], or recent 10-year average PRISM daily data that are calculated on a bi-monthly basis. We consider 10-year average data to be an improvement over 30-year climate normals for producing forecasts because temperatures in CONUS have significantly increased over the past 30 years [24, 25]. The match of mean forecasts of the NMME model's ensemble to the observed value (i.e. skill) varies both spatially and temporally due to topography, season, and the presence of an El Niño-Southern Oscillation (ENSO) signal [26, 27]. It may therefore be more conservative, and provide more consistent predictions, to use 10-year averages instead of NMME data to avoid potential issues with skill. However, we caution that the 10-year average data do not simulate variation in daily $T_{min}$ and $T_{max}$, which may result in the under-prediction of degree-day accumulation in the spring or fall as daily $T_{max}$ only slightly exceeds the lower developmental threshold of a species, or for cooler sites that have temperatures that are often near the threshold. We have also prepared and plan to use the National Weather Service gridded National Digital Forecast Database (NDFD) 7-day forecasts (https://www.weather.gov/mdl/ndfd_info) [28] for use in DDRP.

**Phenology modeling: Species data and parameters.** The life history and behavior of a target species must be considered for appropriateness to model in DDRP. In its current form, the platform can model four separate life stages (the egg, the larva or nymph, the pupa or pre-oviposition, and the adult) plus a separately parameterized overwintering stage. As movement and migration are not handled by DDRP, it is currently limited in its ability to model migratory species, such as those that may establish in southern areas of their potential range and migrate yearly to more northern areas. Species that lack an overwintering stage, which are common in tropical and subtropical areas, may be difficult to model because the timing of first spring activities and stages present cannot be accurately estimated. Currently DDRP is entirely temperature-driven, so species whose growth and reproduction are strongly influenced by additional environmental factors such as day length or moisture may not be accurately modeled.

DDRP requires data on the developmental temperature thresholds (in either degrees Celsius or Fahrenheit) and durations for each life stage of an insect species in degree-days (Fig 1 and Table 1). These data are typically collected in the laboratory by measuring how temperature influences the rate of development, although data derived from season-long monitoring studies are also used [14, 29]. We round Fahrenheit values of thresholds to the nearest integer in all DDRP models because it allows for simpler communication of models to end-users, and it is a long standing convention for degree-day models in the United States. A different developmental threshold may be assigned to each stage, although we typically solve for a common threshold if differences across the stages are minimal. Additionally, applying common thresholds allows a modified version of the DDRP model to be used by the site-based modeling tool at USPEST.ORG (https://uspest.org/dd/model_app), which requires common thresholds across stages, and to more easily compare the DDRP and site-based model implementations for a given species. Presently, the model depends upon a fixed starting date such as January 1, specified by the user for the entire region of interest. The duration of the overwintering stage represents the number of degree-days that must accumulate from the start of the year for the stage to complete.

**Table 1. Species-specific parameters used in DDRP with corresponding values for *Epiphyas postvittana* (light brown apple moth) and *Neoleucinodes elegantalis* (small tomato borer).**

| Parameter | Code | E. postvittana | N. elegantalis |
|---|---|---|---|
| Lower developmental thresholds (˚C) | | | |
| Egg | eggLDT | 7.2 | 8.89 |
| Larvae | larvaeLDT | 7.2 | 8.89 |
| Pupae | pupaeLDT | 7.2 | 8.89 |
| Adult | adultLDT | 7.2 | 8.89 |
| Upper developmental thresholds (˚C) | | | |
| Egg | eggUDT | 31.1 | 30 |
| Larvae | larvaeUDT | 31.1 | 30 |
| Pupae | pupaeUDT | 31.1 | 30 |
| Adult | adultUDT | 31.1 | 30 |
| Stage durations (˚C degree-days) | | | |
| Egg | eggDD | 127 | 86 |
| Larvae | larvaeDD | 408 | 283 |
| Pupae | pupDD | 128 | 203 |
| Adult | adultDD | 71 | 96 |
| Overwintering larvae | OWlarvaeDD | varies | – |
| Overwintering adult | OWadultDD | – | varies |
| Phenological events (˚C degree-days) | | | |
| Overwintering stage event | OWEventDD | varies | varies |
| Egg event | eggEventDD | 126 | 80 |
| Larvae event | larvaeEventDD | 203 | 140 |
| Pupae event | pupaeEventDD | 128 | 100 |
| Adult event | adultEventDD | 22 | 55 |
| Cold stress | | | |
| Cold stress temperature threshold (˚C) | coldstress_threshold | 3 | 6 |
| Cold degree-day (˚C) limit when most individuals die | coldstress_units_max1 | 875 | 1150 |
| Cold degree-day (˚C) limit when all individuals die | coldstress_units_max2 | 1125 | 1600 |
| Heat stress | | | |
| Heat stress temperature threshold (˚C) | heatstress_threshold | 31 | 32 |
| Heat stress degree-day (˚C) limit when most individuals die | heatstress_units_max1 | 375 | 750 |
| Heat stress degree-day (˚C) limit when all individuals die | heatstress_units_max2 | 550 | 1000 |
| Cohorts | | | |
| Degree-days (˚C) to emergence (average) | distro_mean | 210 | 50 |
| Degree-days (˚C) to emergence (variance) | distro_var | 2500 | 1500 |
| Minimum degree-days (˚C) to emergence | xdist1 | 100 | 0 |
| Maximum degree-days (˚C) to emergence | xdist2 | 320 | 111 |
| Shape of the distribution | distro_shape | normal | normal |
| Degree-day calculation method | calctype | triangle | triangle |

Phenology model parameter values were derived from previous studies and an analysis of published data for *E. postvittana* [30–32], and from an analysis of published data for *N. elegantalis* [33]. Climatic suitability model parameter values were estimated by calibrating models in accordance with outputs of a CLIMEX model for each species. For both species, the phenological events for egg, larvae, and adults are beginning of egg hatch, mid-larval development, and first egg laying, respectively. The phenological event for pupae is first adult emergence for *E. postvittana* and mid-pupal development for *N. elegantalis*. The duration and timing of the phenological event for the overwintering stage will vary according to the number of cohorts applied in model runs (see text for details). Both models applied the single triangle method ("triangle") with upper threshold to calculate degree-days.

Users must specify the number of degree-days that are required for the overwintering stage to complete development and emerge for the growing season. These data are typically gathered using field monitoring studies, whereby the temporal distribution of emergence times and number of individuals that emerge on a given date is documented [e.g. 32]. Assigning a single value to the overwintering stage duration parameter would assume that an entire population develops simultaneously, which may not be biologically realistic because several intrinsic (with a genetic basis) and extrinsic (e.g. microclimate, nutrition) factors can produce variation in development rates within a species [19, 34]. Indeed, phenology models that incorporate developmental variability in a population may have increased predictive power [17, 19, 35]. DDRP therefore allows the duration of the overwintering stage to vary across a user-defined number of cohorts (groups of individuals in a population). Much of the intrinsic variability in insect development during a generation often occurs in the overwintering stage [36], although developmental variation may occur in any life stage [17, 37, 38]. DDRP uses five parameters to generate a frequency distribution of emergence times: the mean, variance, low bound, and high bound of emergence times, and the shape of the distribution (Gaussian or lognormal; Table 1). The platform uses these data to estimate the relative size of the population represented by each cohort, which initializes the population distribution that is maintained during subsequent stages and generations. Individuals within each cohort develop in synchrony.

Users may specify the timing (in degree-days) of phenological events that are important to their target system to generate phenological event maps in DDRP, which depict the estimated calendar dates of the event over a time frame of interest. We typically generate phenological event maps based on temperature data for an entire year so that events for multiple generations of each of the five life stages are modeled. For example, phenological event maps that depict when the overwintering stage would emerge may be useful for identifying start dates for surveillance operations for a species, whereas maps for subsequent generations could help with planning operations later in the year. The timing of phenological events may be based on life stage durations (e.g. the end of the egg stage signifies egg hatching), or on occurrences within a stage such as the midpoint or peak of oviposition or adult flight. Currently, one user-defined phenological event for each life stage for up to four generations may be modeled, although the platform could be modified to predict multiple events for each stage (e.g. first, midpoint, and end of the stage) for any number of generations.

**Climatic suitability modeling: Species data and parameters.** Climatic suitability modeling in DDRP is based on cold and heat stress accumulation and requires data on temperature stress threshold and limits of a species (Fig 1 and Table 1). While parameter values may be estimated from laboratory or field experiments, such data are lacking for most species. Additionally, extrapolating laboratory data to the field to project accumulation of stress is difficult due to oversimplification of the number of variables and the temporal and spatial variation in natural environments [39]. We have been using the CLIMEX software [40] (Hearne Scientific Software, Melbourne, Australia), which is one of the most widely used species distribution modeling tools for agricultural, livestock and forestry pests and non-pests [6, 7], to assist with climatic suitability model parameterization in DDRP. Laboratory collected data may help with parameterizing a CLIMEX model; however, model parameters are fine-tuned and the model is fitted using observations from the species' known geographical distribution [40, 41].

DDRP was designed to be complementary to CLIMEX in several ways to facilitate climatic suitability model parameterization, but the two programs also differ in several respects (Table 2). Both platforms use a process-based modeling approach in which parameters that describe the response of a species to temperature stress are included in calculations of climatic suitability. Certain model outputs, particularly maps of temperature stress accumulation, are therefore directly comparable. DDRP uses the stress accumulation method of CLIMEX in

**Table 2. Comparison of the characteristics, parameters, and outputs of climatic suitability models in DDRP and CLIMEX.**

| Attributes | DDRP | CLIMEX |
|---|---|---|
| Temporal range (time frame) | Any–historical, real-time, near forecast, and climate change forecasts | Historical (1961–1990), and future climate change forecasts for 2030, 2050, 2070, 2080, 2090, and 2100 |
| Temporal scale (time step) | 1 day (daily) for PRISM data–others potentially accommodated | Typically weekly values interpolated from monthly data |
| Spatial scale | Any–default is a 2′ resolution (*ca.* 4 km) for PRISM data | CliMond data at a 30′ (*ca.* 55 km at equator) or 10′ (*ca.* 20 km) resolution; others potentially accommodated |
| Factors influencing climatic suitability | Cold and heat stress | Cold, heat, dry, and wet stress plus population growth |
| Modeling process overview | Estimates daily cold and heat stress accumulation and determines whether total accumulations exceed the moderate (max1) or severe (max2) cold and heat stress limits | Estimates weekly population growth and the accumulation of stress (cold, heat, dry, and wet); population growth is reduced when accumulations are too low or too high to maintain metabolism |
| Climate stress parameters | | |
| Temperature stress thresholds | Upper and lower cold and heat stress thresholds in Celsius units | Upper and lower cold and heat stress thresholds in Celsius or degree-day units |
| Temperature stress rates | Cold and heat stress accumulation limits (max1 and max2); stress units accumulate linearly over time (consecutive days not weighted higher than non-consecutive days) | Weekly cold and heat stress accumulation rate based on thresholds in Celsius units (similar to DDRP) or degree-day units; stress units accumulate exponentially over time (consecutive weeks are weighted higher than non-consecutive weeks). |
| Moisture stress thresholds | None | Upper and lower dry and wet stress thresholds |
| Moisture stress rates | None | Weekly dry and wet stress accumulation rate |
| Total no. of parameters possible | 6 | 38 |
| Total no. of parameters typically used | 6 | 21 |
| Depiction of the potential distribution | Areas not under moderate or severe cold and heat stress exclusions | Typically areas with an Ecoclimatic Index $\geq 1$ (the Ecoclimatic Index is calculated using annual growth and stress indices) |
| Outputs | Gridded and graphical outputs of 1) cold and heat stress unit accumulation, and 2) cold, heat, and all (cold plus heat) stress exclusions | Tabular and graphical outputs of 1) cold, heat, dry, and wet stress unit accumulation, and 2) the temperature, moisture, growth, and ecoclimatic index |

For simplicity, we do not show CLIMEX parameters related to interaction stress indices (hot-wet stress, hot-dry stress, cold-wet stress, and cold-dry stress) or to radiation, substrate, light and diapause indices.

which cold and heat stress units begin to accumulate when temperatures exceed the cold and heat stress temperature thresholds, respectively. In both platforms, cold stress units are calculated as the difference between $T_{min}$ and the cold stress temperature threshold, and heat stress units are calculated as the difference between $T_{max}$ and the heat stress temperature threshold. However, DDRP uses daily temperature data and stress accumulates linearly over time, whereas CLIMEX uses average weekly temperature data and stress accumulates at a weekly rate that becomes exponential over time (each week's stress in multiplied by the number of weeks since the stress first exceeded zero) [40]. These differences appear to be minor in using CLIMEX outputs and parameters to support calibration of DDRP parameters, as demonstrated in our case studies (see 'Case Studies'). Similar to CLIMEX, DDRP uses a single cold and heat stress temperature threshold for all life stages, and stress units accumulate across the entire time period of interest (i.e. across all life stages and generations). We apply the cold and heat stress threshold of the life stage that would be most likely to experience the coldest or hottest temperatures of the year, respectively. DDRP assumes that stress could indirectly kill

individuals by restricting their activity, or directly cause mortality through extreme cold or heat events such as a hard freeze.

Importantly, DDRP was designed to model climatic suitability based on daily current or forecast temperature data at fine spatial scales (e.g. a single state or region), which would give users insight into the potential risk of establishment or spread during a particular season or year. In contrast, CLIMEX is normally used to estimate a species' potential distribution using coarse-scale (10′ and 30′ resolution) global gridded 30-year monthly climate normals centered on 1975 (1961–1990) or future projections from selected global circulation models (GCMs) [42]. In theory CLIMEX could be used for real-time climatic suitability, but it has no native ability to import and process common gridded formats and is incapable of using daily resolution climate data. Thus, DDRP's climatic suitability models are intended to improve the efficiency of surveillance and trap deployment at a relatively small focal area for a current or near-future time period, whereas CLIMEX models provide a more general and coarse-scale assessment of suitability based on averaged climate data.

Relying on real-time climatic suitability models for decision support on where to employ pest management and eradication operations for a given year or season is preferable to using models based on 30-year climate normals. A model that uses current climate data is more biologically relevant because the risk of establishment in an area would be affected by the conditions that a species physically experiences, not by averages of historical climate. Additionally, climate in CONUS is changing rapidly, so models based on climate normals may produce unrealistic predictions of present-day climatic suitability. Over the past *ca*. 30 years, the average annual temperature in CONUS has increased by 1.2°F (0.7°C), the number of freezing days has declined, and extreme temperature events have increased in frequency and intensity [24, 25]. Nonetheless, DDRP is not currently capable of including moisture factors in the modeling process like CLIMEX, so model predictions for moisture-sensitive species in very arid or wet areas should be interpreted with caution. We discuss the potential implications of generating a climatic suitability model based solely on temperature in the 'Discussion.'

We compare CLIMEX's predictions of temperature stress accumulation and overall climatic suitability to similar outputs in DDRP to help parameterize a DDRP climatic suitability model. Temperature stress thresholds may be calibrated so that predictions of cold and heat stress accumulation at the end of the year are spatially concordant with CLIMEX's predictions. Climatic suitability in CLIMEX is estimated with the Ecoclimatic Index (EI), which is scaled from 0 to 100, and integrates the Annual Growth Index and the Annual Stress Index (all climate stress indices) to give an overall measure of favorableness of a location or year for long-term occupation by the target species [40, 41]. An EI $\geq$ 1 is often used as a threshold for defining whether a location is suitable for long-term survival, although an EI exceeding 20 or 30 (depending on the species) is sometimes used to indicate a highly suitable climate [40, 41]. As discussed in more detail in 'Case Studies', temperature stress limits in DDRP can be adjusted so that areas predicted to be suitable by CLIMEX are also included in DDRP's prediction of the potential distribution.

Comparing DDRP climatic suitability model outputs to those of CLIMEX for model fitting purposes is naturally more appropriate when temperature data are derived from the same time period. We have therefore been using a PRISM $T_{min}$ and $T_{max}$ 30-year average data set centered on 1975 (1961–1990) to match the time-schedule of the CliMond CM10 (also 1961–1990) world climate data set currently supplied with CLIMEX [42]. We temporally downscaled monthly PRISM estimates for 1961–1990 because DDRP requires daily data and PRISM daily temperature data for years prior to 1980 are not available. For each month of a given year, a bilinear interpolation method was used to assign each day an average temperature value that was iteratively smoothed and then adjusted so that the monthly averages were correct.

## 2) Daily time step

DDRP models insect phenology and climatic suitability by stepping through each day of a specified time period and calculating degree-day and temperature stress accumulation at each grid cell of a focal area (Fig 1). The time period may span the entire year of interest, or include only a subset of days such as those during the growing season. Users may sample and save daily modeling results every 30 days, 14 days, 10 days, seven days, two days, or one day. Results are saved in multi-layer rasters that are processed and analyzed after the daily time step to produce final model outputs. We describe the phenology and climatic suitability modeling process and outputs in more detail in the following sections.

**Phenology model.** DDRP calculates daily degree-days over the specified time period using developmental temperature threshold information and gridded temperature data that have been cropped to the extent of the focal area (Fig 1). Currently DDRP has two methods to calculate degree-days: the simple average using an upper threshold with a horizontal cutoff, and the single triangle method with upper threshold [43–45]. The single triangle method is also used as a close approximation to the more complex sine-curve calculation method [45].

With the exception of phenological event maps, which are computed only for the last day of the daily time step, DDRP saves the following phenology model results for each sampled day:

1. *Accumulated degree-days.* While daily degree-days are calculated for each life stage, the cumulative degree-days are summed only for the first cohort of the larval stage, as these degree-day maps are representative for all cohorts and life stages. Accumulated degree-days calculated for larva will be the same for other life stages if common developmental thresholds are used.

2. *Life stages.* The life stage present (overwintering stage, eggs, larvae, pupae, and adults) for each cohort.

3. *Number of generations.* The current generation count for each cohort. If the model is run for an entire year, then the output for the last day of the year would represent the potential voltinism of the species. The generation count increases when adults progress to the egg stage (i.e. oviposition occurs).

4. *Phenological event maps (optional).* The timing of phenological events is estimated by computing daily degree-day totals from the gridded temperature data, and storing the day of year when an event threshold is reached. Event results are generated only on the last day of the daily time step (typically, the last day of the year) because the entire time period must be analyzed for all potential event days to be considered.

**Climatic suitability model.** Independently from the phenology model simulations, DDRP calculates daily cold and heat stress accumulations and compares these estimates to user-defined moderate and severe stress limits to delineate the potential distribution of the species (Fig 1). We opted to use moderate and severe stress limits to reflect two distinct themes. First, they may provide a way to depict the potential for short term vs. longer term establishment. For most species, the potential distribution could be represented by areas where cold and heat stress have not exceeded the severe or moderate stress limits, as these should allow for long-term survival. DDRP depicts these areas with maps of cold stress exclusion, heat stress exclusion, and all stress exclusion (cold plus heat stress exclusions; Fig 1). Areas under moderate stress exclusion may represent temporary zones of establishment in which a species establishes only during a favorable season, such as after an annual migration event. Conversely, areas under severe stress exclusion do not allow for even short-term establishment. Typically we visualize exclusion maps calculated for the last day of the year (day 365) under investigation

to provide insight into the potential distribution for an entire growing season. For CONUS, the northern range limit is typically delineated by cold stress and the southern range limit, if any, is delineated by heat stress.

Second, using two levels of stress may provide a way to represent uncertainty for estimating the potential distribution. As discussed in more detail in the 'Discussion,' several sources of uncertainty and error in the modeling process may bias model predictions, such as applying inappropriate parameter values, using climate data with low skill or poor spatial resolution, ignoring biotic factors such as species interactions, or ignoring non-temperature abiotic factors such as microclimate effects, moisture, and photoperiod [34, 46]. Defining the potential distribution as areas under severe stress only would typically provide a broader estimate than a definition based on both stress levels. While this approach may over-predict the risk of establishment, conducting surveys over too broad an area is probably better than surveying too small of an area, which may allow a new invasive species to establish and spread.

### 3) Post time step processing

After the daily time step has completed, DDRP combines and analyzes results across cohorts and generates final multi-layer rasters (".tif" GeoTIFF files) and summary maps (".png" image files) for each sampled day. If multiple cohorts were modeled, then DDRP uses estimates of the relative size of the population represented by each cohort to calculate the relative size of the population (totaling 100%) in any given life stage and generation. For phenological event maps, the earliest and average day of year that an event occurs across cohorts is calculated, and these are displayed as calendar dates (month-day) on summary map outputs (Fig 1). DDRP also generates a summary map that depicts the life stages of each generation that are present on a given day. Owing to complexities involved with depicting estimates of the relative size of the population that is represented by each generation and stage, these maps are produced only for the middle cohort because it should represent a large proportion of the population if a normal distribution of emergence times is applied.

DDRP integrates mapping of phenology and climatic suitability so that users can use a single model output to obtain guidance on their "where" and "when" questions (Fig 1). For example, a user involved with planning surveys may want to know where a target species may establish, and within those areas, when populations may emerge from overwintering. Each output of the phenology model with the exception of accumulated degree-days will be associated with two additional outputs for each sampled day (or the last day for a phenological event map): 1) one that includes severe stress exclusion only, and 2) one that includes both severe and moderate stress exclusions. For example, a phenological event map with severe and moderate stress exclusions for 2018 (all 365 days) would present predicted dates of the selected event only in areas where long-term establishment is predicted (Fig 2).

**System and software requirements.**   DDRP requires the R statistical software platform and can be run from the command line or within RStudio [47]. It takes advantage of functions from several R packages for data manipulation, analysis, and post-model processing. The raster package [48] is used to crop daily temperature rasters to the focal area, store and manipulate daily loop raster results, and process and further analyze results for each cohort. Many non-spatial data manipulations are conducted with functions in the dplyr, tidyr, and stringr packages [49–51]. The ggplot2 package [52] is used to generate and save summary maps of raster outputs, and options from the command line argument are parsed using the optparse package [53].

DDRP capitalizes on the multi-processing capabilities of modern servers and computers to run multiple operations in parallel, which is made possible with the R packages parallel and doParallel [54]. This significantly reduces computation times, particularly in cases where

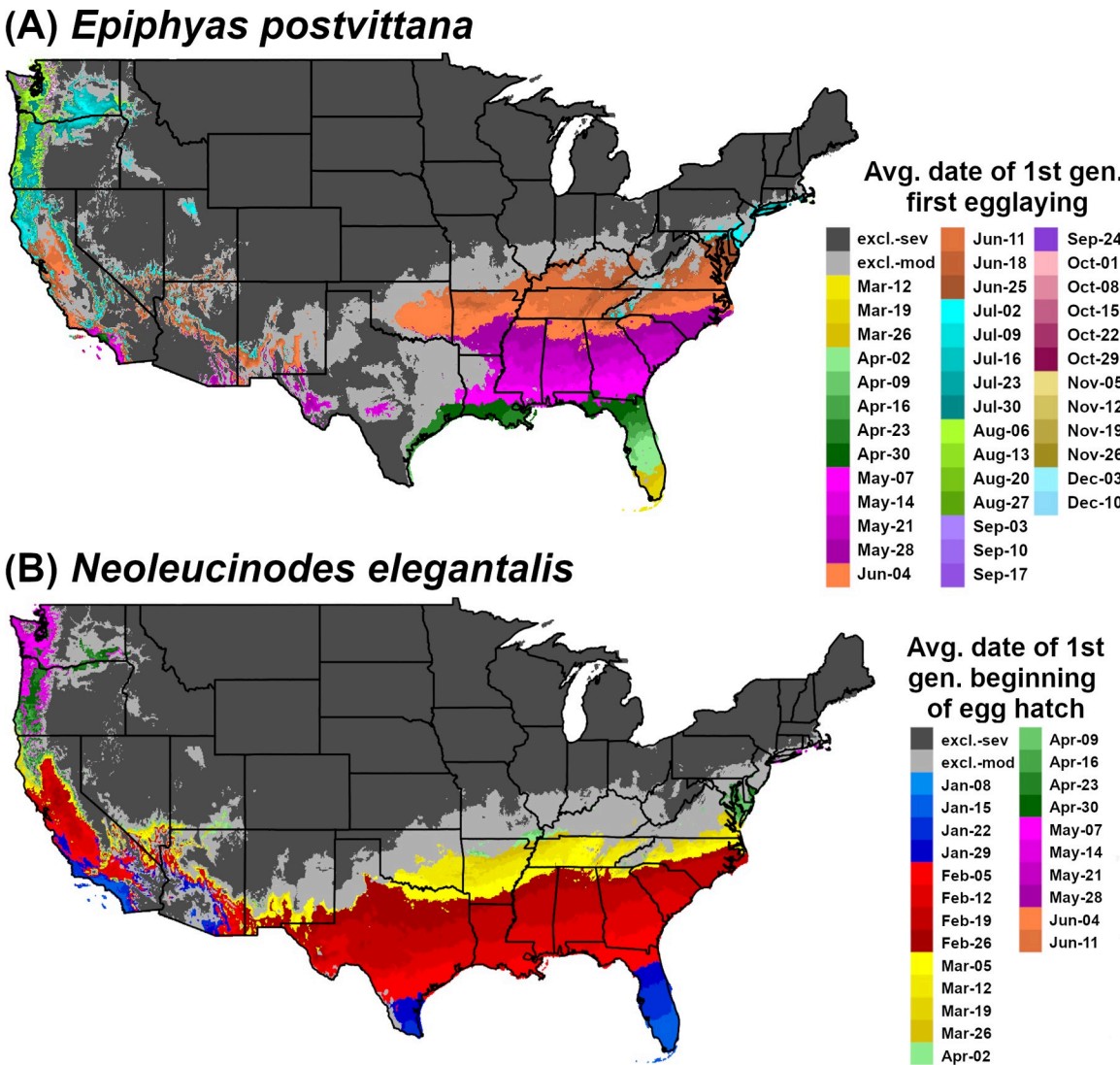

**Fig 2.** Phenological event maps generated by DDRP for (A) *Epiphyas postvittana* (light brown apple moth) and (B) *Neoleucinodes elegantalis* (small tomato borer) in CONUS in 2018. The map for *E. postvittana* shows the average date of egg laying by first generation females, whereas the map for *N. elegantalis* shows the average date of first generation beginning of egg hatch. Both maps include estimates of climatic suitability, where long-term establishment is indicated by areas not under moderate (excl.-moderate) or severe (excl.-severe) climate stress exclusion.

modeling is conducted with multiple cohorts and across large areas. For example, parallel processing is used to crop rasters for multiple dates, run multiple cohorts in the daily time step, and to analyze time step outputs for multiple days or files simultaneously. For very large areas (currently defined as the Eastern United States and CONUS), temperature rasters are split into four tiles and both the tiles and cohorts are run in parallel in the daily time step.

We recommend running DDRP on a server or computer with multicore functionality because certain processes are very memory intensive and may execute slowly or stall without parallel processing. The platform was designed to run on a Linux server, but we have also successfully used it on a Windows OS with eight cores. As an example, a model that was run on a Linux server with 48 cores completed *ca*. 3.5 times faster than it did on a Windows OS with eight cores.

## Case studies

**Climatic suitability, voltinism, and phenological events in *Epiphyas postvittana*.** The light brown apple moth, *E. postvittana* (Walker 1863) (Lepidoptera: Tortricidae), is a leafroller pest native to southeastern Australia, including Tasmania [30]. The species invaded Western Australia, New Zealand, New Caledonia, England, and Hawaii more than 100 years ago [55–57], and has been established in California since 2006 [58, 59]. It poses a significant threat to agricultural production in the United States because it feeds on more than 360 host plants, including economically important fruits such as apple, pear, citrus and grapes [30, 31, 60]. For example, an economic risk analysis of *E. postvittana* to four major fruit crops (apple, grape, orange, and pear) in CONUS estimated an annual mean cost of US$105 million associated with damage to crops and control, quarantine, and research [60]. The CAPS program at APHIS conducts annual surveys for *E. postvittana* at various counties across CONUS.

A summary of phenology and climatic suitability model parameters used for *E. postvittana* in DDRP is reported in Table 1. We assigned all life stages a lower developmental threshold of 7.1°C (45°F) [30] and an upper developmental threshold of 31.1°C (88°F). Laboratory studies revealed small differences in the lower developmental threshold (< 1°C) across different life stages [30, 31]. The upper developmental threshold value is based on studies showing that all life stages cease development between 31–32°C [30–32]. We derived life stage durations (in degree-days Celsius; hereafter, DDC) for *E. postvittana* based on an analysis of published data [30], which resulted in 127, 408, 128, and 71 DDC for eggs, larvae (females on young apple foliage), pupae, and adults to 50% egg laying, respectively (S1 Appendix).

We set the overwintering stage to larva because the predominant overwintering stage of *E. postvittana* in the United States are the late larval instars [61, 62]. We applied seven cohorts to approximate a normal distribution of emergence times that spanned 100 to 320 DDC (average = 210 DDC) based on a report that overwintering larvae at four sites in California required between 102 and 318 DDC to finish development [61]. This would correspond to the time required for mid-stage (3rd–5th instars, average 4th instar) female larval feeding on old foliage (0.45 × 494 DDC = 210 DDC), after a January 1 start date. The single triangle method was used to calculate degree-days.

We tested the hypothesis that DDRP can correctly predict the timing of first spring egg laying and generation length for *E. postvittana* by analyzing three monitoring data sets that were collected in and around the San Francisco Bay Area in California over a 12 year time frame (2008–2009, 2011–2014, and 2019–2020). All three data sets were comprised of moth count data that had been collected via pheromone trap surveys on a bi-weekly or monthly basis by USDA APHIS or the University of California Cooperative Extension. For each data set, we estimated the difference between the date of the peak in first spring flight and DDRP predictions of the average date of first spring egg laying. This event was chosen because peak flight would likely happen at about the same time that peak egg laying is occurring [63]. Additionally, we used DDRP to calculate the number of degree-days that accumulated between the last peak fall flight and first peak spring flight for four winters (2011–2012, 2012–2013, 2013–2014, and 2019–2020), which should serve as a rough estimate of generation time. These estimates were compared to the generation time that is used by the DDRP model (823 DDC for the overwintering generation; 734 DDC are used for later generations that have young foliage to feed upon [30]). Model runs for each year used PRISM data and applied seven cohorts and a normal distribution of emergence times. Additional details about the data sets and methods for this analysis are presented in S2 Appendix.

We generated a CLIMEX model for *E. postvittana* using CLIMEX version 4.0 [40] to help parameterize the climatic suitability model in DDRP. The model applied a combination of

parameter values (Table 3) derived from two previous CLIMEX studies of this species [64, 65]. However, we used a cold stress threshold (TTCS) of 3°C, which is lower than He et al.'s (2012) value (5°C) [64], and higher than Lozier and Mill's (2011) value (1.5°C) [65]. We applied a top-up irrigation (additional simulated rainfall) rate of 2.5 mm day$^{-1}$ for the winter and summer season because irrigation mitigates the hot-dry climate that limits distribution of *E. postvittana* within CLIMEX. We fit the CLIMEX model using a data set of 393 georeferenced locality records from Australia (N = 317) and New Zealand (N = 76), which were obtained from GBIF.org (18th July 2019; GBIF Occurrence Download https://doi.org/10.15468/dl.a4ucei) and Nick Mills at UC Berkeley (pers. comm.).

For model validation, we used locality records from 14,949 sites in California where the California Department of Food and Agriculture detected *E. postvittana* between 2007 and 2013, and from an additional 47 localities in California where it was positively diagnosed between 2011 and 2017. We removed localities that occurred within the same CLIMEX grid cell, which resulted in 140 unique localities. Of these, 95% (133/140) were in areas predicted to have high suitability (EI > 20), 4% (6/140) were in areas predicted to have low suitability (0 < EI < 20), and 0.007% (1/140) were in unsuitable areas (S1 Fig). These findings suggest that the CLIMEX model accurately predicted suitable climates at the majority of localities where the species is known to be established in CONUS, and it is therefore a useful tool for calibrating a climatic suitability model in DDRP.

**Table 3. Parameter values used to produce a CLIMEX model for *Epiphyas postvittana* (light brown apple moth) and *Neoleucinodes elegantalis* (small tomato borer).**

| | | E. postvittana | | | N. elegantalis | |
|---|---|---|---|---|---|---|
| CLIMEX parameter | Code | Lozier & Mills (2011) | He et al. (2012) | This study | da Silva et al. (2018) | This study |
| Temperature | | | | | | |
| Limiting low temperature (°C) | DV0 | 7.5 | 7 | 7 | 8.8 | 8.8 |
| Lower optimal temperature (°C) | DV1 | 15 | 13 | 13 | 15 | 15 |
| Upper optimal temperature (°C) | DV2 | 25 | 23 | 23 | 27 | 27 |
| Limiting high temperature (°C) | DV3 | 31 | 30 | 31 | 30 | 31 |
| Degree-days per generation (°C days) | PDD | 673.6 | 673.6 | 673.6 | 588.2 | 588.2 |
| Moisture | | | | | | |
| Limiting low moisture | SM0 | 0.15 | 0.25 | 0.15 | 0.35 | 0.35 |
| Lower optimal moisture | SM1 | 0.5 | 0.8 | 0.8 | 0.7 | 0.7 |
| Upper optimal moisture | SM2 | 0.8 | 1.5 | 1 | 1.5 | 1.5 |
| Limiting high moisture | SM3 | 1.4 | 2.5 | 2.5 | 2.5 | 2.5 |
| Cold stress | | | | | | |
| Cold stress temperature threshold (°C) | TTCS | 1.5 | 5 | 3 | – | 6 |
| Cold stress temperature rate (week$^{-1}$) | THCS | −0.005 | −0.0005 | −0.0005 | – | −0.0005 |
| Cold stress degree-day threshold (°C days) | DTCS | – | – | – | 15 | – |
| Cold stress degree-day rate (week$^{-1}$) | DHCS | – | – | – | 0.001 | – |
| Heat stress | | | | | | |
| Heat stress temperature threshold (°C) | TTHS | 31 | 31 | 31 | 30 | 31 |
| Heat stress temperature rate (week$^{-1}$) | THHS | 0.0045 | 0.01 | 0.002 | 0.0007 | 0.00084 |
| Dry stress | | | | | | |
| Dry stress threshold | SMDS | 0.15 | 0.2 | 0.15 | 0.35 | 0.35 |
| Dry stress rate (week$^{-1}$) | HDS | −0.005 | −0.01 | −0.005 | −0.001 | −0.001 |
| Wet stress | | | | | | |
| Wet stress threshold | SMWS | 1.4 | 2.5 | 2.5 | 2.5 | 2.5 |
| Wet stress rate (week$^{-1}$) | HWS | 0.001 | 0.002 | 0.001 | 0.002 | 0.002 |

In DDRP, we generated a climatic suitability model for *E. postvittana* using the daily down-scaled PRISM $T_{max}$ and $T_{min}$ estimates for 1961–1990 and calibrated model parameters in accordance with the CLIMEX model (Fig 3). Specifically, we compared maps of temperature stress accumulation, and adjusted temperature stress limits so that most areas predicted to be under moderate and severe climate stress by DDRP had low ($20 > EI > 0$) or zero ($EI = 0$)

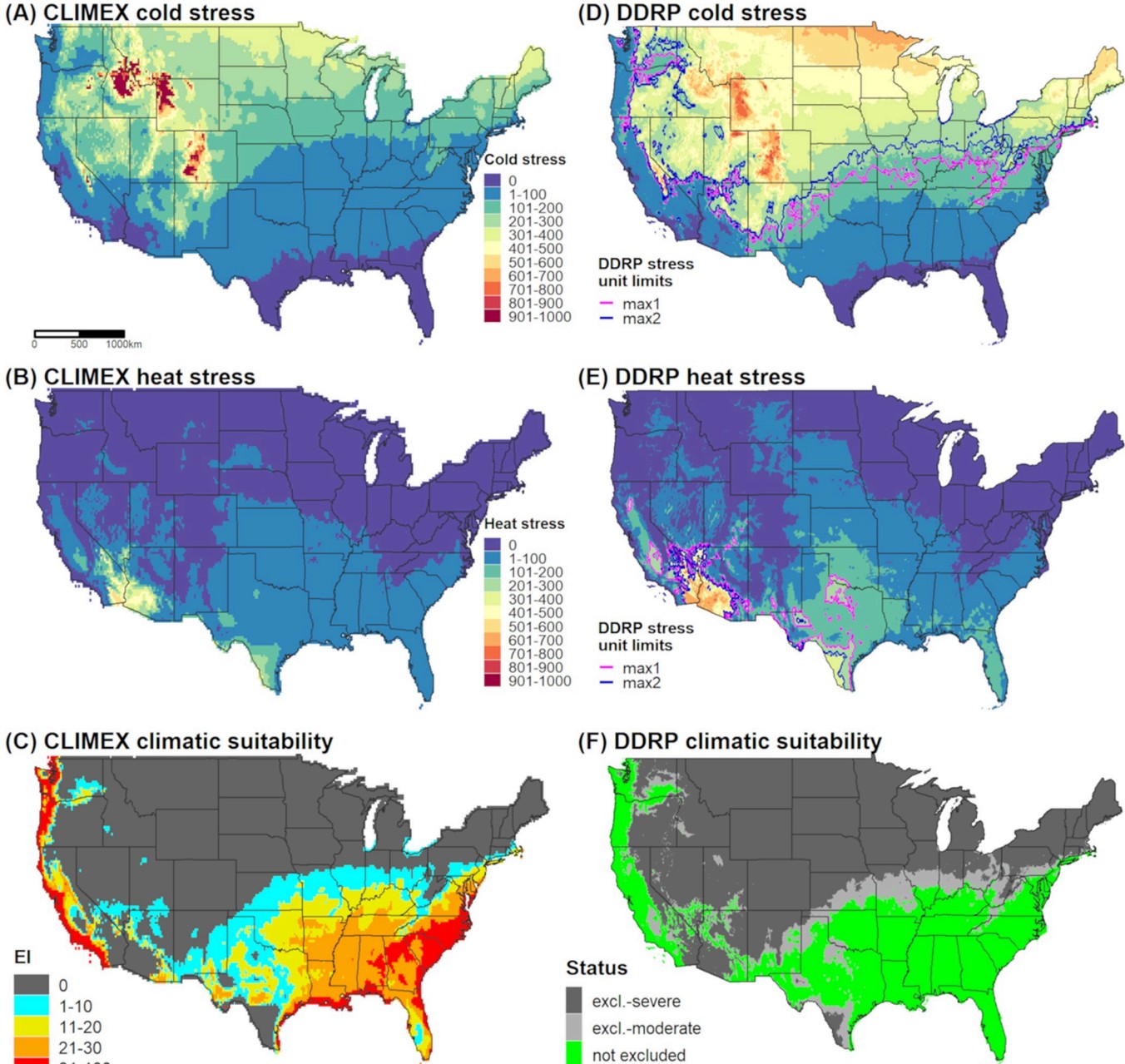

**Fig 3.** Predictions of cold stress, heat stress, and climatic suitability for *Epiphyas postvittana* (light brown apple moth) in CONUS produced by CLIMEX (A–C) and DDRP (D–F) based on 1961–1990 climate normals. Climatic suitability is estimated by the Ecoclimatic Index (EI) in CLIMEX, and by combining cold and heat stress exclusions in DDRP. In DDRP, long-term establishment is indicated by areas not under moderate (excl.-moderate) or severe (excl.-severe) climate stress exclusion. Cold and heat stress units in DDRP were scaled from 0 to 1000 to match the scale in CLIMEX. CLIMEX maps were generated for this study based on parameters documented in Table 3 and have not been previously published.

suitability according to CLIMEX, respectively. Additionally, we modeled climatic suitability for the species in California for each year between 2010 and 2019 and calibrated stress parameters to maximize the fit of the model to 144 georeferenced records that were collected over the same time period (N = 144 from GBIF, N = 77 from Nick Mills). For each modeled time period, we extracted climate stress exclusion values for presence localities in the California Department of Food and Agriculture data set to test our hypothesis that DDRP could correctly predict the potential distribution of the species. We removed localities that occurred within the same 4 km grid cell, which resulted in 872 unique validation localities.

Finally, we modeled phenology and climatic suitability for *E. postvittana* in 2018 to provide insight into its potential voltinism, seasonal activities, and risk of invasion in particularly warm temperatures. The summer of 2018 in the United States was the warmest since 2012 and tied for the fourth-warmest on record (NOAA website https://www.noaa.gov/news/summer-2018-ranked-4th-hottest-on-record-for-us; last accessed 11/21/19). We generated a phenological event map that depicted the date of first egg laying by first generation females, because this activity is relevant to monitoring both eggs and the emergence of adults, which typically occurs two to three days prior to egg laying.

**Climatic suitability, voltinism, and phenological events in *Neoleucinodes elegantalis*.** The small tomato borer, *N. elegantalis* (Guenée) (Lepidoptera: Crambidae), is native to South America and is distributed throughout the Neotropics including in Mexico, Central America, and the Caribbean [66, 67]. A major insect pest of tomato (*Solanum lycopersicum*), it also attacks fruits of other plants belonging to the family Solanaceae including eggplant, paprika, naranjilla, and green and red pepper [67]. There are at least 1175 recorded interceptions of the species from the United States, where it is considered a serious threat to agricultural biosecurity because it lowers tomato production in South America [68]. The CAPS program has conducted surveillance for *N. elegantalis* since at least 2011.

A summary of phenology and climatic suitability model parameters used for *N. elegantalis* in DDRP is reported in Table 1. We re-analyzed data from a laboratory study on the development of *N. elegantalis* on hybrid tomato (Paronset) at five temperatures [33] to estimate a common lower temperature threshold for all life stages, which involved adding a point to force the x-intercept to an integer value in degrees Fahrenheit. We weighted the analysis to select a common lower threshold for immature stages, which are the longest in duration, because this should produce the lowest error for the overall life cycle. The lower threshold values for immature stages were very similar to the overall egg-to-adult value of 8.89˚C (48˚F), so we chose 8.89˚C as the common threshold instead of a higher one solved for the adult pre-oviposition stage (11.5˚C). We estimated the duration for eggs, larvae, pupae, and adults to peak oviposition as 86, 283, 203, and 96 DDC, respectively. This analysis is presented in S3 Appendix.

*Neoleucinodes elegantalis* has no apparent photoperiodic response, diapause, or specific overwintering stage. In subtropical climates in Brazil, the insect remains active throughout the year if host plants are available [33]. We defined adults as the overwintering stage and used January 1 as the model start date for CONUS because few host plants would be available for immature stages at this time. We assumed that adult feeding and host search activities could begin immediately if temperatures are suitable, and that first egg laying would subsequently occur after the estimated pre-oviposition period of *ca.* 55 DDC. The durations of later events (first to peak oviposition, immature development, etc.) were estimated from previously published data [33, 69]. We applied seven cohorts to approximate a normal distribution of emergence times that spanned 0 to 111 DDC (average = 50 DDC) because overwintered adults begin finding hosts over this time frame, and we used the single triangle method to calculate degree-days. Unfortunately, we were unable to find monitoring data that were suitable for validating the DDRP phenology model for *N. elegantalis*. Population monitoring studies of the

species have been conducted in Brazil; however, the species at these locations does not have a discrete overwintering stage so peaks in flight are more or less sporadic and not indicative of first spring activity or voltinism.

We used two different approaches to parameterize a DDRP climatic suitability model for *N. elegantalis* for CONUS. First, we used a similar approach taken for *E. postvittana* in using a CLIMEX model to help calibrate temperature stress parameters in DDRP. As detailed in S4 Appendix, we re-parameterized a previously published CLIMEX model for *N. elegantalis* [70] because it appeared to underpredict suitability in warmer areas where the species is known to occur. Briefly, this involved fitting a CLIMEX model using 228 locality records from the Neotropics and validating it using a separate set of 54 localities (S1 Table). CLIMEX correctly predicted suitable conditions (EI $\geq$ 1) at 96% of the validation localities (51/53) that fell within a unique grid cell (S2 Fig), which suggests that the model is robust at predicting climatic suitability. We therefore compared DDRP maps of temperature stress accumulation to those of CLIMEX, and adjusted temperature stress limits so that most areas predicted to be suitable by CLIMEX were also included in the potential distribution by DDRP (Fig 4). We considered any area that had an EI $\geq$ 1 to potentially be suitable because some localities occurred in areas that had only marginal suitability (S2 Fig).

We further calibrated the DDRP climatic suitability model for *N. elegantalis* by fitting a model for Brazil using 83 locality records and daily gridded $T_{min}$ and $T_{max}$ data for 2005 to 2016 at a 0.25˚ (*ca.* 28 km) spatial resolution [71]. To our knowledge, Brazil is the only country within the range of *N. elegantalis* for which daily gridded meteorological data are available. We did not use temperature data for earlier years (1980 to 2004) because they had lower accuracy [71]. A random subsample of 70% of locality records (S1 Table) were used to fit the DDRP model (N = 58), and the remaining 30% (N = 25) were used for model validation. We modeled climatic suitability for each year between 2005 and 2016 and iteratively adjusted heat stress parameters to maximize model fit. Brazil represents a relatively warm part of the distribution of *N. elegantalis*, so we were unable to calibrate cold stress parameters. During this process, we also considered how well DDRP predictions of suitability and heat stress aligned with those of CLIMEX (Fig 5). Despite DDRP's use of climate data for different time periods than CLIMEX for this analysis (i.e. individual recent years vs. 30-year climate normals), its predictions of heat stress accumulations and climate stress exclusions over several years were often spatially concordant with CLIMEX's estimates (Fig 5). This finding provided some additional reassurance beyond the validation analysis that DDRP could correctly model climatic suitability of the species.

Finally, we modeled phenology and climatic suitability in DDRP using temperature data for 2018. We generated a phenological event map for the average date of the beginning of egg hatch of the first generation (Fig 2B). Predictions of egg hatch could enhance population control of *N. elegantalis* because this species is most vulnerable to pesticides before larvae enter the fruit of host plants [72].

## Results

DDRP predictions of the timing of first spring egg laying for *E. postvittana* were consistent with estimates of first spring peak flight derived from three population monitoring data sets for most years. Predicted dates of first spring egg laying differed by fewer than six days from the peak date for 2013, 2014, 2019, and 2020, with some occurring later than the peak (2014, 2020) and others occurring earlier (2013, 2019; S2 Table). Model prediction error was larger for 2012 (*ca.* 3 weeks later than peak spring flight). For 2008 and 2009, DDRP predictions for three of the four analyzed counties were consistent with the observed month of peak spring flight. Estimates of degree-day accumulation between the last peak fall and first spring flights

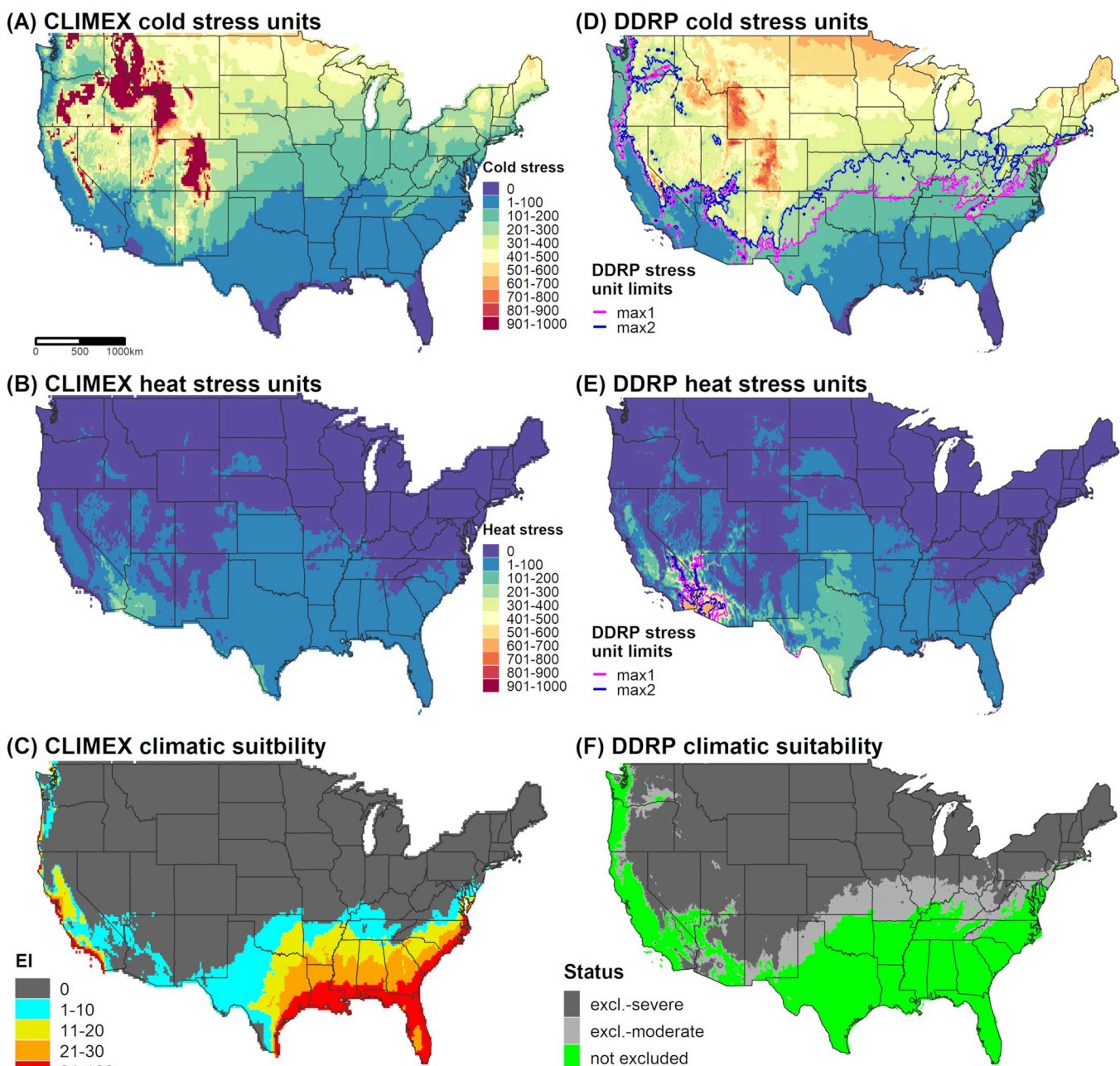

**Fig 4.** Predictions of cold stress, heat stress, and climatic suitability for *Neoleucinodes elegantalis* (small tomato borer) in CONUS produced by CLIMEX (A–C) and DDRP (D–F) based on 1961–1990 climate normals. Climatic suitability is estimated by the Ecoclimatic Index (EI) in CLIMEX, and by combining cold and heat stress exclusions in DDRP. In DDRP, long-term establishment is indicated by areas not under moderate (excl.-moderate) or severe (excl.-severe) climate stress exclusion. Cold and heat stress units in DDRP were scaled from 0 to 1000 to match the scale in CLIMEX. CLIMEX maps were generated for this study based on parameters documented in Table 3 and have not been previously published.

across four winters (S3 Table) were similar to or slightly lower (average = 722 DDC, range = 693–827 DDC) than the generation time assumed by the DDRP model (822 DDC).

For all modeled years, DDRP predictions of the potential distribution of *E. postvittana* were in agreement with the vast majority of validation localities. The model based on 30-year climate normals predicted suitable conditions at 99.7% (871/874) of validation localities. On

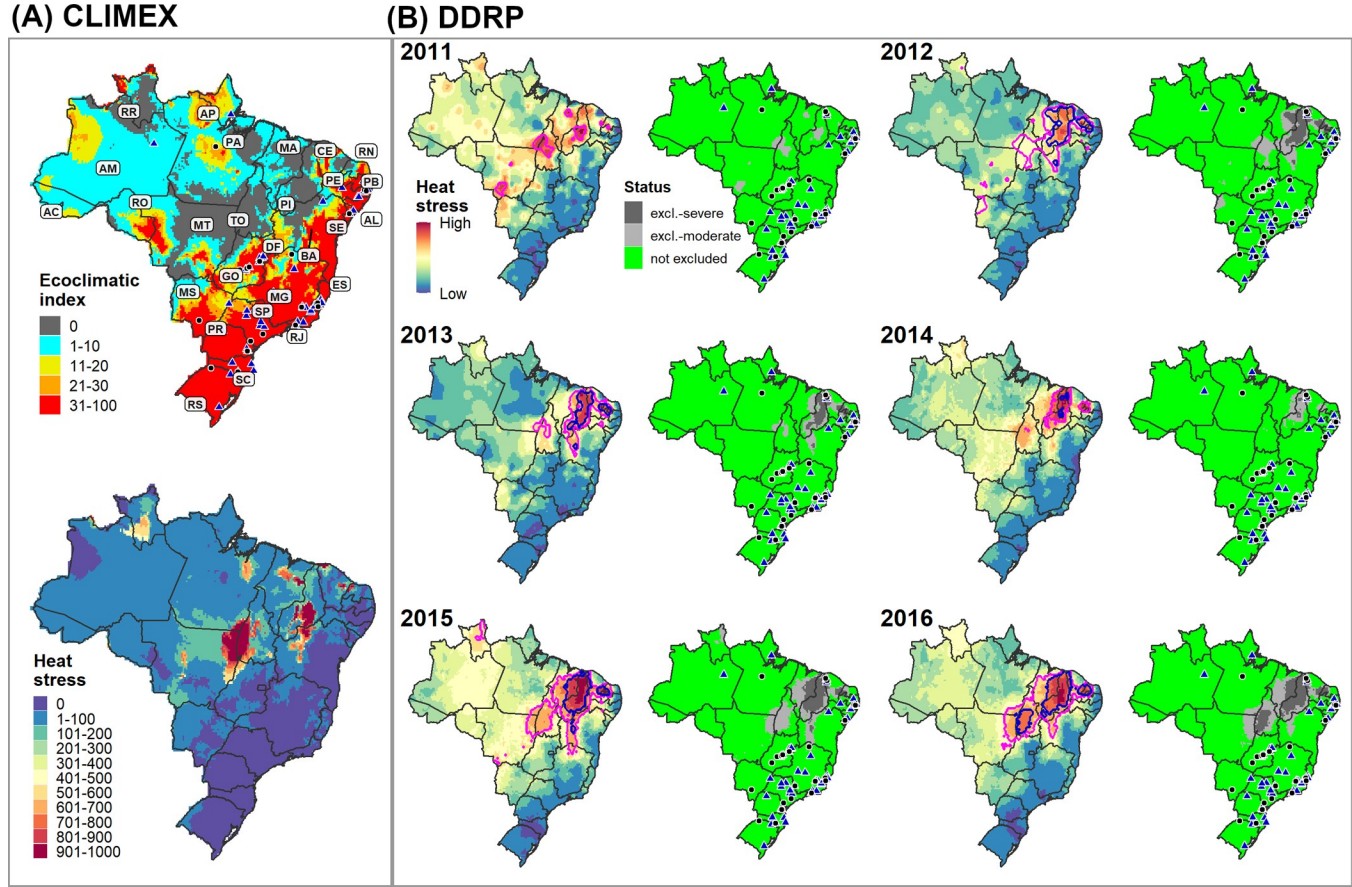

**Fig 5.** CLIMEX (A) and DDRP (B) predictions of heat stress and climatic suitability for *Neoleucinodes elegantalis* in Brazil. The CLIMEX model is based on 1961–1990 climate normals whereas DDRP models are presented for all years between 2011 and 2016. Locality records that were used to fit (blue triangles) and validate (black circles) the DDRP climatic suitability model are depicted. Both models predicted suitable conditions in regions where *N. elegantalis* is widespread, including in the states of Espírito Santo (ES), Goiás (GA), Minas Gerais (MG), Pernambuco (PE), Rio de Janeiro (RJ), Paraná (PR), Rio Grande do Sul (RS), Santa Catarina (SC), and São Paulo (SP). The pink and blue lines in DDRP heat stress maps depict the moderate and severe temperature stress limits, respectively.

average, models based on climate data for each year between 2010 and 2019 predicted suitable conditions at 97.4% (range = 92.4–99.4%) of validation localities, whereas only 2.3% (range = 0.6–6.8%) and 0.4% (range = 0.2–0.8%) were predicted to be excluded by moderate and severe heat stress, respectively.

Similarly, DDRP models for *N. elegantalis* in Brazil predicted suitable conditions at most validation locations where the species is known to occur (Fig 5B). On average, models based on climate data for each year between 2005 and 2016 predicted suitable conditions at 92.4% (range = 87–100%) of validation localities. Conversely, 4.7% (range = 0–13%) and 2.9% (range = 0–8.7%) were predicted to be excluded by moderate and severe heat stress, respectively. Two localities which were incorrectly excluded by heat stress in 13 of the 15 modeled years occurred in the Serra de Ibiapaba, which is a high-elevation region (maximum of 850 m) in the state of Ceará where elevations change quickly across relatively short distances (e.g. *ca.* 30 km in many areas). The coarse spatial resolution of input climate data (*ca.* 28 km$^2$) relative to the scale of these elevational changes most likely explains inaccurate predictions. For relatively warm years since 2011, heat stress was predicted to exclude *N. elegantalis* from warmer regions of Brazil (e.g. in the states of Maranhão, Piauí, and Tocantins; Fig 5B) where the species has not been documented to our knowledge.

Cold stress was the major determinant of the potential distribution of *E. postvittana* and *N. elegantalis* in CONUS according to DDRP analyses based on 30-year climate normals (1961–1990). Both species were excluded from the northern half of CONUS by cold stress, with the exception of (mostly) western parts of Oregon and Washington (Figs 3D and 4D). Heat stress excluded *E. postvittana* from most of southern Arizona and California, and in parts of Texas (Fig 3E). Conversely, *N. elegantalis* was excluded only from the hottest parts of southern Arizona and California (Fig 4E). When considering both cold and heat stress exclusions, the potential distribution of both species included western parts of the Pacific states (California, Oregon, and Washington), the Southeast, and southern parts of the Northeast (in Delaware, Maryland, and New Jersey). The predicted northern range limit of *E. postvittana* extended farther north than *N. elegantalis* and included parts of some Midwestern states (Kansas, Missouri, Illinois, and Indiana).

DDRP predicted a smaller potential distribution for *E. postvittana* and *N. elegantalis* in 2018 compared to 1961–1990 (Figs 2 and 6). According to model runs for 2018, *E. postvittana* was excluded by heat stress from warm areas of CONUS that were included in its potential distribution under historical conditions, including parts of Arizona, New Mexico, Texas, and the Central Valley of California (S3 Fig). Conversely, the more heat tolerant *N. elegantalis* experienced only small reductions in its potential distribution in southeastern California and southwestern Arizona due to heat stress (S3 Fig). In the East, the northern range limit for each species was predicted to be slightly farther south in 2018 than in 1961–1990 due to higher levels of cold stress (S3 Fig).

Predictions of potential dates for phenological events and voltinism for *E. postvittana* and *N. elegantalis* in 2018 varied substantially by latitude in eastern CONUS (Figs 2 and 6). The earliest date of egglaying for the first generation of *E. postvittana* was predicted to be as early as mid-March in Florida to as late as December in the Pacific Northwest (Fig 2A). The average date of first generation beginning of egg hatch for *N. elegantalis* was predicted to begin in the first week of January in Florida but not until mid-June in the Pacific Northwest (Fig 2B). Thus, the timing of monitoring trap installation to detect ovipositing adults and eggs of *E. postvittana*, or larvae of *N. elegantalis*, could vary widely across CONUS. For both species, DDRP predicted as many as seven to nine generations in coastal areas of the Southeast, compared to only one or two generations in colder parts of the Pacific Northwest (Fig 6). Three to six generations were predicted for most other regions of CONUS. These findings may indicate that the Southeast would experience the longest duration of pest pressure.

## Discussion

### DDRP as a decision support tool

An improved understanding of *where* an invasive species could potentially establish as well as *when* developmental stages are expected to occur have the potential to support and dramatically improve strategic and tactical pest management decisions [6–8]. DDRP is a new spatial modeling platform that integrates real-time and forecast predictions of phenology and climatic suitability (risk of establishment) of invasive insect pests in CONUS, providing insights into both where and when to focus detection efforts for a given year or growing season. These predictions may help with detecting the presence of invasive species in the shortest time possible after they arrive and reproduce, which increases the chance of eradication success and makes other rapid response measures (e.g. deployment of biological control) possible and less costly [4]. For example, phenological event maps for *E. postvittana* and *N. elegantalis* (Fig 2) identify high-risk areas where surveillance activities could be concentrated, in addition to providing estimated dates of activities that can ensure timely trap placement. Thus, users can use a single program to address decision support needs for early pest detection.

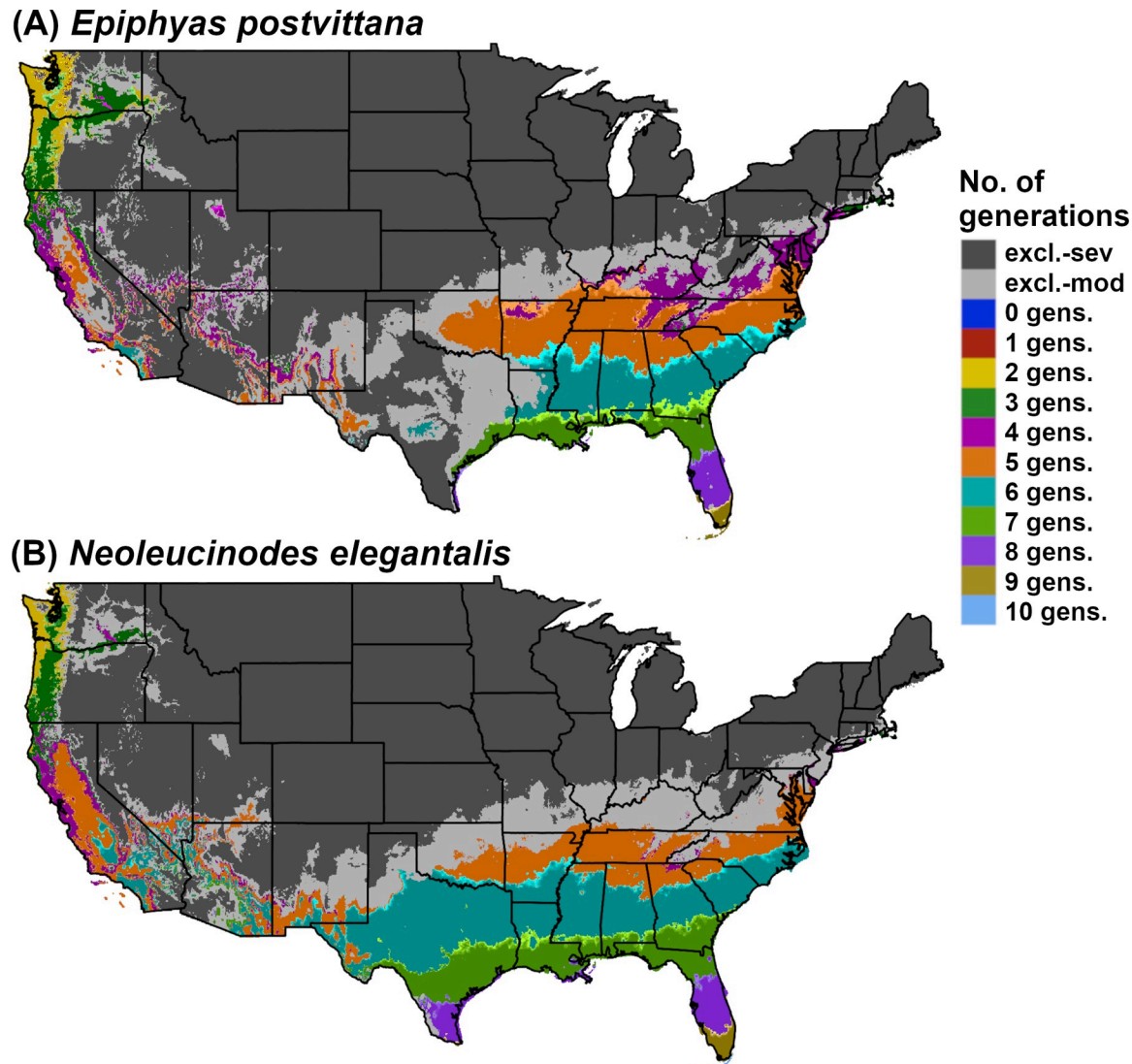

**Fig 6.** DDRP model predictions of voltinism (number of generations per year) in (A) *Epiphyas postvittana* (light brown apple moth) and (B) *Neoleucinodes elegantalis* (small tomato borer) in CONUS for 2018. Maps include estimates of climatic suitability, where long-term establishment is indicated by areas not under moderate (excl-mod) or severe (excl-sev) climate stress exclusion.

Additionally, DDRP may be a useful decision support tool for monitoring and managing populations of IPM pests and classical biological control agents. For example, growers have used predictions of the timing of first egg hatch for codling moth [*Cydia pomonella* (Linnaeus)], first emergence of western cherry fruit fly [*Rhagoletis indifferens* (Curran)], and first spring oviposition of spotted wing drosophila [*Drosophila suzukii* (Matsumura)] to help monitor and reduce populations of these major crop pests [73–76]. Phenology models for biological control insects can help managers schedule sampling trips to coincide with insect presence on the target organism, and to estimate the synchrony of insect and host phenology at a given location [77, 78]. DDRP's estimates of voltinism may provide insights into expected pressure on target organisms, as higher voltinism should translate to greater agent population growth and biocontrol success.

Our hypothesis that DDRP can correctly predict the timing of first spring egg laying and the generation length of *E. postvittana* was generally supported, although there was variation in

prediction error across years and regions. DDRP predictions were averaged across several grid cells for two of the three monitoring data sets because they lacked precise location information, which would likely contribute to prediction errors. Additionally, errors in predictions of spring egg laying may reflect either differences in the effects of evening temperatures and wind speeds on flight activity [79], differences in diet which affects development rates [80], or a lack of consistent correspondence between male flight and peak egglaying, albeit the two events are often found to correlate well enough at least during the springtime to use as a biofix for models informing treatment decisions. Most estimates of generation length that we obtained by measuring degree-day accumulation for the overwintering generation across four years were higher than the value used by the DDRP phenology model, which may in part be due to the model's assumption that overwintering larvae feed on old apple leaves, which slows development [30].

DDRP predictions of the potential distribution of *E. postvittana* and *N. elegantalis* in simulations for multiple recent years were in agreement with the vast majority of validation localities, a finding which supports our hypothesis that DDRP can correctly predict the known distribution of each species. Our models for 2018 indicated that heat stress excluded populations of both species from a greater area of the Southwest compared to 1961–1990, which is consistent with studies showing that global warming may reduce species' distributions in warmer parts of their range [81]. However, determining whether these putative range shifts are persistent or temporary would require combining model runs for multiple recent years or seasons. For example, trends in the geographic distribution of climate stress exclusions over several years or seasons could be visualized with a probability surface map. Estimating the direction of range shifts may also provide insights into the response of the species to future climate change. Nonetheless, the differences that we documented in predictions of climatic suitability based on climate data for 1961–1990 compared to 2018 suggests that an area's contemporary risk of establishment is different than it was *ca*. 30 years ago. DDRP's ability to produce climatic suitability models in real-time may provide more meaningful insights into areas that are presently suitable for an invasive species, and therefore allow for more effective placement of surveillance operations. Additionally, the ability to capture interannual variability of pest risk in space-time may allow decision-makers in pest management to react more adaptively to risk.

There are numerous opportunities for improving and extending the applications of DDRP. The platform can be readily modified to use daily temperature data for a different region than CONUS, as we demonstrated in our analysis of climatic suitability for *N. elegantalis* in Brazil. Additionally, DDRP could be modified to model other types of temperature-dependent organisms such as non-insect invertebrates, plants, plant-pathogenic bacteria and fungi, and insect plant and animal virus vectors. The platform has been tested for and could be used through an on-line web interface, although there is the potential that memory intensive processes could overload a server host. This issue, as yet untested, could be addressed by using a cloud computing platform. We describe some additional features that could be added to potentially improve model accuracy and expand the list of outputs in more detail below. The most recent code for DDRP is available at GitHub (https://github.com/bbarker505/ddrp_v2.git), where we invite scientists and practitioners to jointly develop the platform and database of species models.

## Comparison of DDRP to other platforms

From a historical perspective, DDRP could be considered a partial descendant and spatialized version of the PETE (Predictive Extension Timing Estimator) phenological modeling platform that was established as a standard in the mid-1970s [82]. Features in common include a cohort approach to population phenological modeling using daily degree-days as the main input, provision for major insect life stages and a separately parameterized overwintering distribution,

an open-source non-proprietary standard for sharing, and a focus on agricultural extension (applied decision support). Unlike PETE, DDRP is spatialized and therefore able to produce a variety of mapping outputs including phenological event maps, and it also includes options to use separate thresholds for different life stages and to generate climatic suitability models. DDRP could be improved by adding certain features of the PETE platform including the use of a diapause trigger, and a distributed delay function that would allow population spread to increase with each subsequent generation.

Web-hosted daily degree-day maps and degree-day lookup table maps for pest management are available from several platforms including USPEST.ORG, SAFARIS, and Enviroweather, but none of these are capable of predicting climatic suitability like DDRP. Additionally, degree-day lookup tables have certain underlying assumptions which may result in an oversimplified model that is lacking in biological realism, such as assuming that multiple species or life stages within a species have a common lower temperature threshold, and that early season development in a population begins at the same time (i.e. no developmental variation). In contrast, DDRP uses species specific parameters including stage-specific lower and upper developmental temperature thresholds, and it can account for developmental variation within populations by generating and combining results across multiple cohorts that complete the overwintering stage at different times. Compared to a simple model based on generation time degree-days, a well-parameterized DDRP model would likely produce more accurate predictions of voltinism and spring activity because different temperature thresholds for multiple life stages may be used, the overwintering stage is parameterized separately from the post-winter stage (e.g. overwintering adult vs. adult), and the timing of spring activity is summarized across multiple cohorts.

The Insect Life Cycle Modelling (ILCYM) [83, 84] and phenModel [85] software packages are open-source R programs that use life stage specific parameters to model temperature-based insect phenology, and both incorporate parameters that describe developmental variation within a population. ILCYM is similar to DDRP in its ability to predict phenology and climate based establishment risk in a spatial context. However, ILCYM is a full population dynamics modeling platform that requires life table data at constant and variable temperatures, which are seldom available for anticipated but not yet present invasive insect species. Additionally, published ILCYM risk model simulations use climate normals or GCMs at global or regional scales [83, 86], whereas DDRP was designed to use real-time and forecast climate data for within-season decision support in CONUS. The phenModel package is not spatialized nor capable of modeling climatic suitability, and it would need to be modified to use for an insect species other than the blue willow beetle *Phratora vulgatissima* [85].

## Uncertainties, limitations, and other considerations

**Linear (degree-day) modeling.** DDRP uses a relatively simple degree-day modeling approach, whereas some platforms including ILCYM, phenModel, and devRate [87] offer complex functions to model nonlinear responses of insects to temperature. Degree-day models are ideal for multi-species platforms like DDRP because there are sufficient data to parameterize a degree-day model for most insect pest species of economic importance in the United States [18, 88]. Linear degree-day models are also readily calibrated and sometimes constructed entirely using field data, making them more practical for extension and decision support use [14]. Additionally, degree-day models require only daily $T_{min}$ and $T_{max}$ data (as opposed to hourly data for most nonlinear models), which are available at a high spatial resolution for CONUS from multiple sources including PRISM and RTMA. Nonetheless, it is important for users to recognize potential sources of error and lack of precision in degree-day models, such as their limited ability to accurately model development at supra-optimal temperatures [13, 14, 89].

**Environmental inputs.** DDRP is intentionally parameterized in a simple, conservative manner, which will hopefully achieve the goal of a parsimonious balance of both model simplicity and accuracy [14, 90]. Nonetheless, DDRP is driven entirely by temperature, and therefore ignores other factors that may affect the development and distribution of insects such as photoperiod, moisture, dispersal, resources, disturbance, and biotic interactions [7, 91]. The potential consequences of this limitation will depend on the biology of the organism under study. For example, dry stress is a major factor restricting the current distribution of *N. elegantalis* in its native range [92–94], and it limits the distribution of *E. postvittana* both in its native range [30, 95] and in Southern California and Arizona [96]. Our CLIMEX model for *N. elegantalis* predicted higher dry stress in arid parts of the Southwest than the rest of CONUS (S4 Fig), which suggests that an absence of moisture factors in DDRP may result in over-predictions of climatic suitability in this region. However, this conservative-leaning error may in fact better reflect human manipulation of the landscape (e.g. greenhouse and irrigation usage) that may allow the species to exist in such regions. Future versions of DDRP that can process gridded moisture data and incorporate moisture stress factors into climatic suitability models may help overcome our current limitations in matching CLIMEX models, and may improve predictions for moisture-sensitive species such as *N. elegantalis*. Additionally, we are developing a version of DDRP that incorporates photoperiodically induced life history events such as winter diapause and summer aestivation, which builds on earlier phenology modeling work that estimated voltinism of photoperiod-sensitive insects [97].

**Presumptive models.** Uncertainties regarding the accuracy of temporal or spatial predictions of invasive species that are not yet established is inevitable, in part because no validation data are yet available, and species interceptions do not imply establishment [7, 98]. DDRP models for species for which only presumptive models exist should therefore be used conservatively. For example, surveillance or management actions could be implemented in advance of predicted phenological events as a precautionary measure (e.g. installing traps even earlier than estimates for the earliest date of overwintering adult emergence). To potentially avoid under-predicting the risk of establishment, the potential distribution could be defined as areas not under severe climate stress as opposed to defining it using both stress levels. Additionally, climatic suitability models generated by DDRP could be combined with those produced using different modeling methods (e.g. correlative, semi-mechanistic, or mechanistic) to create a "hybrid" model, which may increase the reliability of predictions [7, 91].

Web platforms that support sharing of pest observations from the United States will be valuable resources for validating and increasing the predictive performance of DDRP models for species that are already established in CONUS. For example, the iPiPE and its sister platforms (http://www.ipipe.org, https://ipmpipe.org) have created a national information technology infrastructure for sharing pest observations in near real-time and contributing them to a national repository [99]. Similarly, the USA National Phenology Network provides a repository of plant and insect phenology observations contributed by citizen scientists [16]. The National Agricultural Pest Information System (NAPIS; https://napis.ceris.purdue.edu/home) currently has over 5.17 million records from pest detection surveys, and is another potential source of validation data.

**Geographic variation.** Populations of an invading species may exhibit geographic variation in temperature dependent development if genetically divergent individuals are introduced to different areas or rapid evolutionary changes occur in new environments [100, 101]. If the geographic distribution of variation in a relevant thermal trait is well understood, then model accuracy may be improved by building separate models for each genotype. For example, an egg hatch phenology model for a subspecies of the Asian gypsy moth, *Lymantria dispar asiatica* (Vnukovskij), had reduced error compared to a similar model constructed for the European

subspecies that has invaded North America, *Lymantria dispar dispar* (Linnaeus), which has a markedly different predominant phenotype [102]. An alternative approach may be to run several models, each with a different value for the parameter of interest, and present a range of model predictions. Conversely, DDRP could be modified to accept a grid of parameter values so that geographic variation would be accounted for in a single model run.

A lack of knowledge on how early-season environmental conditions or events that initiate the first spring activity of a species (biofix) vary geographically may be a source of error because the model start date affects all downstream predictions. For example, how does first spring activity vary across the wide range of warming conditions possibly encountered for a large region such as CONUS? As a case in point, our phenology model for *N. elegantalis* assumes that moths have only 55 DDC before egg laying behaviors may occur. This assumption may not be valid for sub-tropical zones of the United States, where flight and reproduction could occur even earlier. Conversely, a much longer spring warm-up may be needed in temperate zones because commercial tomatoes are transplanted much later in the year. Studying how first spring activity (adult flight) in *N. elegantalis* potentially varies geographically in Central or South America would help to refine a range of model start times. The phenology model for DDRP could then be parameterized using a necessarily conservative selection of start dates or by inputting a grid of start dates. Using a broad distribution of emergence times to initiate the cohorts could be another approach to accommodate uncertainty in first spring activity.

**Distributed delay.** There is currently no distributed delay function in DDRP, meaning that the overlap in generations and life stages of cohorts does not increase over multiple generations. This is particularly an issue for species that have significant overlap in generations because they continue to develop throughout winter months or lack a temperature or photoperiodic event that synchronize populations, such as *E. postvittana* [31, 57, 61]. DDRP may accurately predict peak events in each generation for these species, but inaccurately predict the first appearance of one or more life stages after the first or second generations because of increasing overlap in generational cohorts. Consequently, phenological event maps should be most reliable for the first few generations. This will be among the high priority issues in development of future versions of the platform.

## Conclusion

DDRP is a new multi-species modeling platform that can integrate mapping of phenology and climatic suitability in real-time to provide timely and comprehensive guidance for stakeholders needing to know both where and when an invasive insect species could potentially invade. When used for surveillance, the platform will hopefully increase chances for early detection of new or spreading invasive threats in the United States, and therefore help pest management programs mitigate their potential damage to agricultural and environmental resources. Additionally, DDRP may help plan monitoring and management efforts for IPM pests and biological control insects, and to predict pest pressure on host plants.

The case studies we presented provided examples of how DDRP models may be parameterized and then run to produce various outputs including gridded and graphical predictions of the number of generations, life stages present, dates of phenological events, and areas of climatic suitability based on two levels of climate stress. We encourage users of DDRP to consider the limitations of the platform, to report the conditions that their model was designed to work under (e.g. a particular region, life stage event, or generation), and to document any known sources of model error that could not be accounted for when providing validation and other feedback reports. Additionally, models should be validated whenever possible, as we did for the *E. postvittana* and *N. elegantalis* models presented in this study. The flexible and open-source

nature of DDRP will facilitate making modifications and improvements, such as adding new environmental factors, using it for other regions besides CONUS, modeling non-insect organisms, expanding the types of model products, and adding features to improve model accuracy.

## Supporting information

**S1 Appendix. Estimating phenology model parameters for *Epiphyas postvittana*.**
(PDF)

**S2 Appendix. Data sources for validating a DDRP phenology model for *Epiphyas postvittana*.**
(PDF)

**S3 Appendix. Estimating a common lower temperature threshold and other phenology model parameters for *Neoleucinodes elegantalis*.**
(PDF)

**S4 Appendix. Methods for fitting and validating a CLIMEX model for *Neoleucinodes elegantalis*.**
(PDF)

**S1 Fig.** Predictions of climatic suitability for Epiphyas postvittana in Australia, New Zealand, and California based on 1961–1990 climate normals according to (A, B) CLIMEX and (C) DDRP (California only).
(PDF)

**S2 Fig. CLIMEX model for *Neoleucinodes elegantalis* in the Neotropics.**
(PDF)

**S3 Fig.** DDRP predictions of cold and heat stress for (A) Epiphyas postvittana and (B) Neoleucinodes elegantalis for 2018.
(PDF)

**S4 Fig. CLIMEX predictions of dry stress for *Neoleucinodes elegantalis* in CONUS.**
(PDF)

**S1 Table. Locality records used for validating the CLIMEX and DDRP climatic suitability models for *Neoleucinodes elegantalis*.**
(PDF)

**S2 Table. Comparison of DDRP predictions for the dates of first spring egg laying ("predicted") to the month (data set 1) or dates (data sets 2 and 3) of peak spring adult flight for *Epiphyas postvittana* in California according to three monitoring data sets ("observed").**
(PDF)

**S3 Table. DDRP predictions of the number of degree-days Celsius (DDC) that accumulated between peaks in last fall flight and first spring flight of *Epiphyas postvittana* in California.**
(PDF)

## Acknowledgments

We extend our thanks to Dan Upper for providing spatial weather data processing and systems administration for the project, Darren Kriticos for feedback and literature on the use of CLIMEX, Peter McEvoy for providing helpful comments and edits to earlier drafts of this

manuscript, Nick Mills for providing locality records for *E. postvittana*, Greg Simmons for providing trapping data that we used to validate DDRP predictions for *E. postvittana*, and Ricardo Silva for providing locality records for *N. elegantalis* in South America.

## Author Contributions

**Conceptualization:** Brittany S. Barker, Leonard Coop, Tyson Wepprich, Fritzi Grevstad, Gericke Cook.

**Data curation:** Brittany S. Barker, Leonard Coop.

**Formal analysis:** Brittany S. Barker, Leonard Coop, Tyson Wepprich.

**Funding acquisition:** Leonard Coop, Fritzi Grevstad, Gericke Cook.

**Investigation:** Brittany S. Barker, Leonard Coop.

**Methodology:** Brittany S. Barker, Leonard Coop, Tyson Wepprich, Gericke Cook.

**Project administration:** Brittany S. Barker, Leonard Coop.

**Resources:** Leonard Coop.

**Software:** Brittany S. Barker, Leonard Coop, Tyson Wepprich, Gericke Cook.

**Supervision:** Leonard Coop.

**Visualization:** Brittany S. Barker.

**Writing – original draft:** Brittany S. Barker.

**Writing – review & editing:** Brittany S. Barker, Leonard Coop, Tyson Wepprich, Fritzi Grevstad, Gericke Cook.

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
