## [Decision Letter · Decision Letter 0]

17 Jul 2020

PONE-D-20-15142

DDRP: real-time phenology and climatic suitability modeling of invasive insects

PLOS ONE

Dear Dr. Barker,

Thank you for submitting your manuscript to PLOS ONE. After careful consideration, we feel that it has merit but does not fully meet PLOS ONE’s publication criteria as it currently stands. Therefore, we invite you to submit a revised version of the manuscript that addresses the points raised during the review process.

The reviewers and I agree that this a well-written manuscript. I appreciate your attention to detail and your thorough explanations of the choices you made in developing DDRP. However, Reviewer 1 expressed reservations about the lack of underlying hypotheses and validation, and I share some of those concerns. As you are probably aware, PLOS ONE has a specific set of criteria for manuscripts that present new software. These criteria relate to utility, validation, and availability (https://journals.plos.org/plosone/s/submission-guidelines#loc-methods-software-databases-and-tools). With respect to validation, “the submitted manuscript must demonstrate that the new tool is an improvement over existing options in some way. This requirement may be met by including a proof-of-principle experiment or analysis; if this is not possible, a discussion of the possible applications and some preliminary analysis may be sufficient.” I believe your goal is to see wide uptake of DDRP, which means that you must meet a fairly high standard for validation that goes beyond preliminary analysis. Certainly, your discussion of the practical advantages of DDRP over alternatives is compelling, but it would be much better if you provided quantitative evidence.

I encourage you to look to the comments of Reviewer 1 for potential opportunities to evaluate model performance. For instance, it should be relatively straightforward to implement (holdout) validation of the predicted distributions for your two example cases and derive some estimates of the error rates. Additionally, you should be able to devise some sort of test of the relative merit of your degree-day modeling method, even with limited data.

We look forward to receiving your revised manuscript.

Kind regards,

Frank H. Koch, PhD

Academic Editor

PLOS ONE

Journal Requirements:

2.We note that [Figure(s) S1, S2, S4, 2, 3, 4 and 5] in your submission contain [map/satellite] images which may be copyrighted. All PLOS content is published under the Creative Commons Attribution License (CC BY 4.0), which means that the manuscript, images, and Supporting Information files will be freely available online, and any third party is permitted to access, download, copy, distribute, and use these materials in any way, even commercially, with proper attribution. For these reasons, we cannot publish previously copyrighted maps or satellite images created using proprietary data, such as Google software (Google Maps, Street View, and Earth). For more information, see our copyright guidelines: http://journals.plos.org/plosone/s/licenses-and-copyright.

1.    You may seek permission from the original copyright holder of Figure(s) [S1, S2, S4, 2, 3, 4 and 5] to publish the content specifically under the CC BY 4.0 license. 

Additional Editor Comments (if provided):

Minor comments:

Table 1 – references/citations for these parameter values? (I realize they’re provided later but it would be good to include references in the caption for readers)

Line 215 – delete “estimates of” (redundant)

Line 271- “high” instead of “highly”

Reviewers' comments:

Reviewer's Responses to Questions

**Comments to the Author**

1. Is the manuscript technically sound, and do the data support the conclusions?

Reviewer #1: No

Reviewer #2: Yes

2. Has the statistical analysis been performed appropriately and rigorously? 

Reviewer #1: No

Reviewer #2: N/A

3. Have the authors made all data underlying the findings in their manuscript fully available?

Reviewer #1: Yes

Reviewer #2: Yes

4. Is the manuscript presented in an intelligible fashion and written in standard English?

Reviewer #1: Yes

Reviewer #2: Yes

5. Review Comments to the Author

Reviewer #1: Barker et al. provide a well written accounting of DDRP, a new software package to calculate degree days and climatic suitability for insects. The authors are correct that refinements are needed to prepare forecasts of the potential distribution and activity of insects over time. Such forecasts could be useful to refine surveys, particularly for invasive species whose complete current distribution may not be known with confidence.

While I do not find any errors, per se, I do not find any underlying hypotheses to underpin the research. One set of hypotheses could be related to the ability of the model to reproduce the distribution or activity of the light brown apple moth or the small tomato borer, the example cases that were provided. Ideally, these forecasts would be for areas that were not used for model development. Such validation is routine for these types of models and is a bare minimum. The model might have high commission error rates, classifying more areas as suitable than exist. It is not clear how well the model performs with respect to seasonal activity/phenology.

The authors discuss at great length different methods of estimating degree days. An argument is made that the method of calculation could result in quantitatively and qualitatively different forecasts. Such a suggestion is intriguing (and is certainly the subject of several other research papers), but the authors provide no new evidence (e.g., trap catch data over time) to demonstrate that the method they selected was substantially better, worse, or no different from other methods.

Another set of hypotheses could be related to the marginal improvement in model forecasts over existing models. As the authors correctly note, a number of platforms exist to provide national degree days based on recent weather. (I was a bit surprised to not see any mention of the Spatial Analytic Framework for Advance Risk Information System [SAFARIS] also sponsored by USDA APHIS.) The authors suggest that one distinguishing feature of the current model is the ability to account for variation in response to degree day accumulations. Yet, no data are provided to measure how the forecast is improved when such variability is added.

The authors description of the method of calculating cold and heat stress is vague. Do they use the same fundamental formulas as are used in CLIMEX. Why? Have they independently vetted those indices to determine how well they perform? Further, why do stress indices need to be determined on a daily basis? Again, with no formal comparisons, I am unclear what new insights the model is providing over existing models.

This model was developed with support from the Cooperative Agricultural Pest Survey. Can the authors describe how surveys were conducted before the development of this platform? How have they changed in response? Is there any evidence that trapping is now more efficient?

Current conclusions are not based on the results of the study. They are primarily aspirational.

Reviewer #2: This article describes a new a new multi-species spatial modeling platform that integrates mapping of phenology and climatic suitability in real-time which has applications for invasive pest detection and management. While many insect phenology modeling platforms are currently available, none to my knowledge, consider that multiple insects cohorts emerging at different times.

Overall, the manuscript is well written. The authors provide good justification for their modeling parameters and adequately discuss the limitations of DDRP. The authors develop two solid insect pest model examples in DDRP which highlight the platform’s products and potential as a decision support tool. The modeling is sound, and the R code is readily available online. There is very little that I can fault here. This article is certainly suitable for publication in Plos One.

General Comments:

The manuscript is a bit on the long side. I would recommend some trimming in the discussion section.

Line by Line comments:

143: Replace “ have simulated” with “simulate”

171: Replace “develop” with “development”

226: Replace “is” with “in”

400-405: DDRP seems like it would have a steep learning curve given that it runs best on a Linux OS and requires knowledge of R. How are these forecasting maps going to made available to pest managers and growers planning their pest management schedule for the growing season?

6. PLOS authors have the option to publish the peer review history of their article (what does this mean?). If published, this will include your full peer review and any attached files.

Reviewer #1: No

Reviewer #2: No

---

## [Author Response · Author response to Decision Letter 0]

11 Nov 2020

Please see the "Response to Reviewers" document

---

## [Decision Letter · Decision Letter 1]

26 Nov 2020

PONE-D-20-15142R1

DDRP: real-time phenology and climatic suitability modeling of invasive insects

PLOS ONE

Dear Dr. Barker,

Thank you for submitting your manuscript to PLOS ONE. After careful consideration, we feel that it has merit but does not fully meet PLOS ONE’s publication criteria as it currently stands. Therefore, we invite you to submit a revised version of the manuscript that addresses the points raised during the review process.

From the Academic Editor: Thank you for working to address the concerns raised during the first round of reviews. In particular, I believe your manuscript is stronger now that it includes and reports on some validation procedures. Reviewer #3 (added for the second review round) provided a handful of relatively minor comments and suggestions. If you address these, the manuscript should be suitable for publication.

We look forward to receiving your revised manuscript.

Kind regards,

Frank H. Koch, PhD

Academic Editor

PLOS ONE

Reviewers' comments:

Reviewer's Responses to Questions

**Comments to the Author**

1. If the authors have adequately addressed your comments raised in a previous round of review and you feel that this manuscript is now acceptable for publication, you may indicate that here to bypass the “Comments to the Author” section, enter your conflict of interest statement in the “Confidential to Editor” section, and submit your "Accept" recommendation.

Reviewer #2: All comments have been addressed

Reviewer #3: (No Response)

2. Is the manuscript technically sound, and do the data support the conclusions?

Reviewer #2: Yes

Reviewer #3: Yes

3. Has the statistical analysis been performed appropriately and rigorously? 

Reviewer #2: (No Response)

Reviewer #3: Yes

4. Have the authors made all data underlying the findings in their manuscript fully available?

Reviewer #2: Yes

Reviewer #3: Yes

5. Is the manuscript presented in an intelligible fashion and written in standard English?

Reviewer #2: Yes

Reviewer #3: Yes

6. Review Comments to the Author

Reviewer #2: (No Response)

Reviewer #3: Overall

This is a very thorough, clear description of a new approach for modeling both species phenology and climate suitability for insect pests. I have only a handful of suggestions for improvements.

Specific things

Abstract

- L3 – “multi-species” is a bit confusing here – prior to reading the whole paper, it’s unclear whether this means that the system can handle multiple species at once (+interactions among them?) or rather that the system is species-agnostic

- L12 – “products” – maybe “outputs” works better here?

- L14-15 – the CLIMEX models bit here is hard for a reader unfamiliar with this to grasp in an Abstract (without the more complete description and explanation provided in the text). Maybe this sentence could be something like “Inputted species parameter values can be derived from laboratory or field studies or established through an additional modeling step.”

Main text

- L200 – “stadium” – is this a typo? Just checking – I don’t have an entomology background

- L440 – this mention of uspest.org seems out of the blue. USPEST is mentioned once early in the paper (L55), though it’s not made clear that the authors run this program or how it intersects with the DDRP package described in this manuscript.

- L457-459 – what is the source of these data? How were they collected? Just a few brief, key details would suffice and be helpful here

- L539 – why did you force Fahrenheit to integer values?

- L856 – “depository” (twice) – maybe “repository” works better?

7. PLOS authors have the option to publish the peer review history of their article (what does this mean?). If published, this will include your full peer review and any attached files.

Reviewer #2: No

Reviewer #3: No

---

## [Author Response · Author response to Decision Letter 1]

29 Nov 2020

Response to Reviewers

PONE-D-20-15142R1

DDRP: real-time phenology and climatic suitability modeling of invasive insects

PLOS ONE

Reviewer #3

This is a very thorough, clear description of a new approach for modeling both species phenology and climate suitability for insect pests. I have only a handful of suggestions for improvements.

Specific things

Abstract

- L3 – “multi-species” is a bit confusing here – prior to reading the whole paper, it’s unclear whether this means that the system can handle multiple species at once (+interactions among them?) or rather that the system is species-agnostic

We made this correction. We added a sentence into the Introduction that provides more context as to how DDRP is a “multi-species” modeling tool.

- L12 – “products” – maybe “outputs” works better here?

We made this correction. For consistency, we changed the word “product” to “output” in other parts of the manuscript as well.

- L14-15 – the CLIMEX models bit here is hard for a reader unfamiliar with this to grasp in an Abstract (without the more complete description and explanation provided in the text). Maybe this sentence could be something like “Inputted species parameter values can be derived from laboratory or field studies or established through an additional modeling step.”

We made this correction except that we used the word “estimated” instead of “established”.

Main text

- L200 – “stadium” – is this a typo? Just checking – I don’t have an entomology background

Technically a stadium is the period of time between two successive molts. We replaced this word with “stage” because the two words may be used synonymously (Schaefer, CW. 1971. Bull. Entomol. Soc. Am. 17:315). 

- L440 – this mention of uspest.org seems out of the blue. USPEST is mentioned once early in the paper (L55), though it’s not made clear that the authors run this program or how it intersects with the DDRP package described in this manuscript.

We agree and have moved this information to the part of the manuscript that presents the input data and parameters in DDRP (second paragraph of “Phenology modeling: species and parameters”). We also clarified the relationship between DDRP and USPEST.ORG. 

- L457-459 – what is the source of these data? How were they collected? Just a few brief, key details would suffice and be helpful here

We added a sentence that summarizes that sources of the data sets and how they were collected. However, the details of the data sets are presented in S2 Appendix to reduce the length of the manuscript.

- L539 – why did you force Fahrenheit to integer values?

Resolving Fahrenheit values to integers in degree-day models is a long standing convention in the U.S. because it allows for simpler communication of models to end-users. We have added this information to the manuscript. 

- L856 – “depository” (twice) – maybe “repository” works better?

Good point – we made this correction.

---

## [Editor Report · Decision Letter 2]

2 Dec 2020

DDRP: real-time phenology and climatic suitability modeling of invasive insects

PONE-D-20-15142R2

Dear Dr. Barker,

We’re pleased to inform you that your manuscript has been judged scientifically suitable for publication and will be formally accepted for publication once it meets all outstanding technical requirements.

Kind regards,

Frank H. Koch, PhD

Academic Editor

PLOS ONE

Additional Editor Comments (optional):

Thank you for completing a final set of revisions. I appreciate your efforts to address all reviewer comments in this and the previous rounds of review.
---

## [Editor Report · Acceptance letter]

21 Dec 2020

PONE-D-20-15142R2 

DDRP: real-time phenology and climatic suitability modeling of invasive insects 

Dear Dr. Barker:

I'm pleased to inform you that your manuscript has been deemed suitable for publication in PLOS ONE. Congratulations! Your manuscript is now with our production department. 

Kind regards, 

on behalf of

Dr. Frank H. Koch 

Academic Editor

PLOS ONE